# Real-time non-contact cellular imaging and angiography of human cornea and limbus with common-path full-field/SD OCT

Viacheslav Mazlin [1✉], Peng Xiao [1,2✉], Jules Scholler [1], Kristina Irsch[3,4], Kate Grieve[3,4], Mathias Fink [1] & A. Claude Boccara[1✉]

In today's clinics, a cell-resolution view of the cornea can be achieved only with a confocal microscope (IVCM) in contact with the eye. Here, we present a common-path full-field/spectral-domain OCT microscope (FF/SD OCT), which enables cell-detail imaging of the entire ocular surface in humans (central and peripheral cornea, limbus, sclera, tear film) without contact and in real-time. Real-time performance is achieved through rapid axial eye tracking and simultaneous defocusing correction. Images contain cells and nerves, which can be quantified over a millimetric field-of-view, beyond the capability of IVCM and conventional OCT. In the limbus, palisades of Vogt, vessels, and blood flow can be resolved with high contrast without contrast agent injection. The fast imaging speed of 275 frames/s (0.6 billion pixels/s) allows direct monitoring of blood flow dynamics, enabling creation of high-resolution velocity maps. Tear flow velocity and evaporation time can be measured without fluorescein administration.

[1] Institut Langevin, ESPCI Paris, PSL University, CNRS, 1 Rue Jussieu, 75005 Paris, France. [2] State Key Laboratory of Ophthalmology, Zhongshan Ophthalmic Center, Sun Yat-sen University, 510060 Guangzhou, China. [3] Vision Institute/CIC 1423, Sorbonne University, UMR_S 968/INSERM, U968/CNRS, UMR_7210, 17 Rue Moreau, 75012 Paris, France. [4] Quinze-Vingts National Eye Hospital, 28 Rue de Charenton, 75012 Paris, France. ✉email: mazlin.slava@gmail.com; xiaopengaddis@hotmail.com; claude.boccara@espci.fr

The cornea is the front part of the eye acting as a clear window into the world. In order to maintain optical clarity, it has a complex and highly-specialized micromorphology of fibers, cells and nerves. The tear film protecting the cornea on top and limbus with blood flow supply at the periphery also play essential roles in maintaining corneal health. A small malfunction in any part of this sophisticated system may lead to a broad range of potentially blinding (4th leading cause of blindness worldwide[1]) corneal disorders: degenerative (keratoconus), inherited or infectious (bacterial, viral, and fungal keratitis). Taking into account that the largest corneal blindness burden falls on developing countries[2], cost-effective disease prevention through early diagnosis and treatment, and through public health programs is preferable over costly surgical interventions[3]. However, early and precise diagnosis is frequently complicated as various pathologies, requiring different therapies, may show the same symptoms on a macroscopic level[4]. Clinical OCT[5–7] with microscopic axial resolution proved to be useful for differentiating between many anterior eye conditions. Nevertheless, a number of pathologies were left unaddressed by clinical OCT, because of an unmet need for high lateral resolution in the en face view. The first screening method to identify specific disease biomarkers with isotropic micrometer-level resolution was in vivo confocal microscopy (IVCM)[8]. Today IVCM is used in clinical practice as an important quantitative tool[9]. Nevertheless, its use is frequently avoided, primarily because of the requirement for direct physical contact with the patient's eye preceded by ocular anesthesia. This results in discomfort for the patient, increased risk of corneal damage and risk of infection. Moreover, IVCM provides a limited field of view (FOV), well below 0.5 mm × 0.5 mm, which results in a long examination time as the clinician searches for the region of interest. Despite its drawbacks, IVCM, has remained the only cell-resolution corneal screening tool available in clinics, with no alternatives up to now.

Recently, we developed in vivo full-field optical coherence tomography (FFOCT) and demonstrated its capability to capture images of the central human cornea, resolving features such as nerves, cells, and nuclei without touching the eye[10]. This technology, originating from the lower speed ex vivo FFOCT[11–14], uses a 2D camera to acquire high-resolution en face images (i) directly without beam scanning artifacts, (ii) rapidly (at a camera frame rate), (iii) with flexible FOV. These qualities distinguish FFOCT from another high-resolution modality—UHR-OCT, which also achieved impressive cell-detail imaging in the cornea and limbus using a conventional scanning approach[15,16]. The combination of benefits, provided by FFOCT (non-contact operation, cell-resolution, and en face optical sectioning without beam scanning artifacts) was achieved thanks to our full-field interferometry approach. Nevertheless, the first in vivo FFOCT design captured an image only when the optical path lengths of the two interferometer arms were perfectly matched, which occurred only when the cornea happened to land in the perfect position. Indeed, given that the in vivo cornea is constantly moving, even during steady fixation, such matching only occurs at rare random moments, prohibiting consistent real-time imaging and visualization of the whole breadth and depth of the cornea, necessary for clinical use of FFOCT.

Here we demonstrate a combined common-path FFOCT/SDOCT microscope (FF/SD OCT), which tracks the axial position of the eye and matches the optical arm lengths of FFOCT in real time to allow consistent imaging of the entire ocular surface (central, peripheral and limbal cornea, limbus, sclera, tear film). We demonstrate that real-time, millimeter-field movies of central and peripheral in vivo corneas consistently reveal cells and nerves, which can be quantified according to the existing medical protocols used for IVCM. This makes FF/SD OCT ready for clinical research and translation into clinical practice as a non-contact OCT-based alternative to IVCM with higher resolution than conventional OCT, and capable of detecting changes in the entire ocular surface. Moreover, we show that our setup also provides the possibility of monitoring tear film evolution, opening a door to quantitative and non-contact assessment of dry eye conditions. Beyond the cornea, the instrument visualizes the scleral and limbal regions, important to stem cell storage and regeneration. FF/SD OCT can also directly view blood flow with high contrast and without fluorescein injection at a high frame rate of 275 frames per second (fps), which, paired with its high-resolution capabilities, allows individual blood cells to be followed. Furthermore, we demonstrate a method to reveal high-resolution blood flow velocity and orientation maps, which may lead the way to new localized diagnostic methodologies of scleral inflammation and approaches to monitor therapeutic effects locally.

## Results

**Tracking eye position with common-path FF/SD OCT**. The central idea in a common-path FF/SD OCT interferometer is to detect the axial position of the cornea in real-time with SDOCT and use this information to optically match the arms of the FFOCT interferometer by moving the reference arm, leading to consistent FFOCT imaging of a moving in vivo cornea (Fig. 1 and Supplementary Movies 1, 2). SDOCT, coupled to the microscope objectives, displays the locations of the corneal surface and other reflecting surfaces in the XZ plane with high axial resolution (<3.9 μm) by acquiring 2D cross-sectional 1.25 mm × 2.7 mm images of backscattered light intensity. 2D SDOCT images are formed through rapid 100 kHz A-line scanning with a galvanometer mirror. Combined SDOCT and FFOCT share the same optical paths in the arms of the FFOCT interferometer, which is apparent in the SDOCT image by appearance of a common-path FF/SD OCT peak in addition to the conventional peaks from the cornea and reference mirror of FFOCT (Fig. 2a–d). As the common-path and reference mirror peaks never overlap, we can simultaneously detect them and calculate actual position of the cornea, imaging depth inside the cornea, optimal reference position, actual reference position and error for validating and improving the feedback loop (Fig. 2e and Supplementary Movie 3). Every 8.2 ± 0.5 ms (mean ± standard deviation (s.d.)) information about the current corneal position is sent to the FFOCT system, where it is used to correct optical mismatch between the interferometer arms.

**Real-time optical path length matching of interferometer arms**. The FFOCT interferometer, equipped with a near-infrared (NIR) 850 nm incoherent light-emitting diode (LED) source and moderate numerical aperture (0.3 NA) 10× air microscope objectives (MO), acquires 2D en face 1.25 mm × 1.25 mm images of XY corneal sections with 1.7 μm lateral and 7.7 μm axial resolutions (see Methods section) by a time-domain two-phase shifting scheme[12] (Fig. 1 and Supplementary Movies 1, 2). Each FFOCT image, composed of 1440 × 1440 pixels, is captured by a 2D CMOS camera in 3.5 ms at 275 FFOCT frames/s (or 0.6 billion pixels/s), which is 130 times faster (in terms of pixel rate) than the state-of-the-art corneal confocal scanning IVCM systems, imaging at 30 fps over 384 × 384 pixels FOV (or 3.6 million pixels/s)[17].

When the FFOCT interferometer is focused on the surface of the cornea, the sample and reference arms are optically matched (or more precisely, the focus spot of light on the cornea matches with the coherence plane, which is located at the position that matches (in terms of optical path length) with the reference mirror). However, as the cornea shifts axially, interferometer

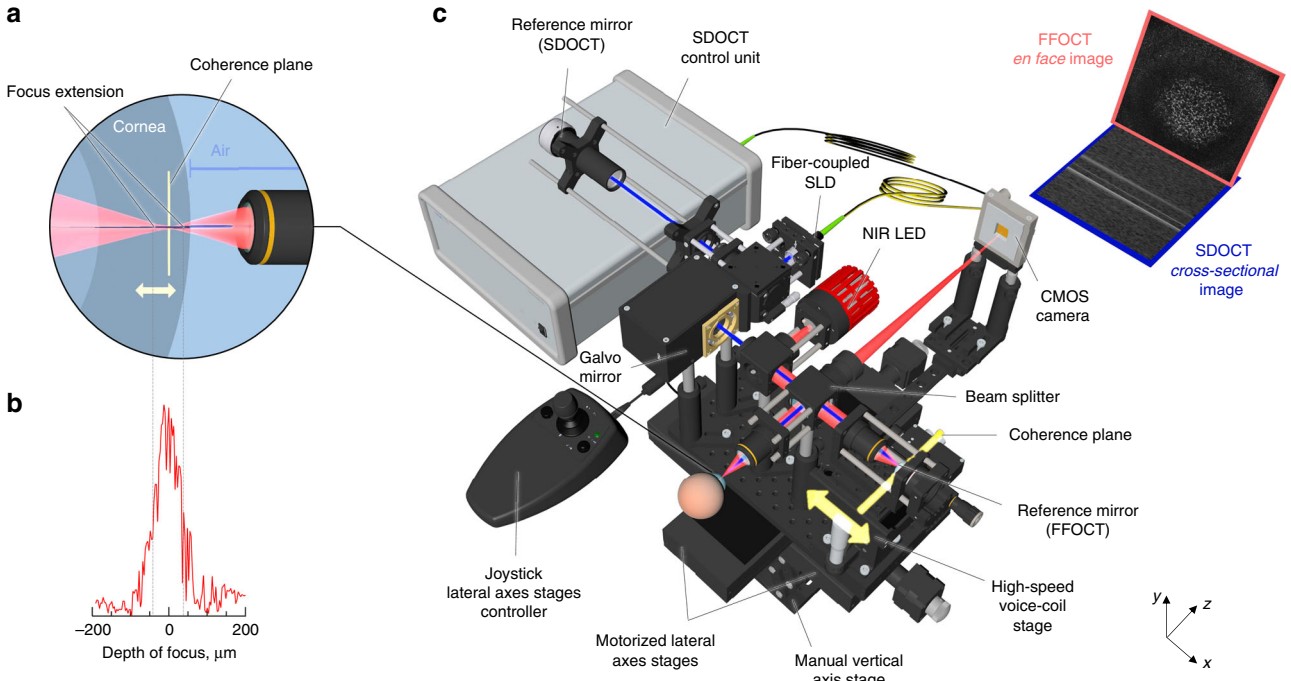

**Fig. 1 Common-path FF/SD OCT with axial eye tracking and real-time defocus adjustment. a** Side-view of the FFOCT sample arm with the cornea. Location of the focus changes with changing corneal position, due to refraction at the air-cornea boundary (Supplementary Movie 1). The location of the coherence plane (depicted in yellow), corresponding to the position of the FFOCT reference arm, also shifts, but in the opposite direction, leading to an optical mismatch between the two and loss of FFOCT signal. **b** Measured depth of focus. **c** The microscope consists of two interferometers: FFOCT, which acquires en face images using a 850 nm light-emitting diode illumination (depicted in red), and SDOCT, capturing cross-sectional images with a 930 nm superluminescent diode light (depicted in blue). SDOCT data is used to calculate the current corneal position and the optical mismatch correction required, which is fed into the fast voice coil stage in the FFOCT reference arm. The stage shifts rapidly to place the coherence plane within the changing position of the depth of focus. As a result, the FFOCT interferometer arms match, and en face images are acquired consistently and in real-time (Supplementary Movies 1, 2). All unlabeled scale bars are 400 μm. The source data underlying the plot in **b** are provided as a Source Data file.

arms become mismatched due to two factors: (i) the spread of the focal point into the sample due to Snell's law, originating from the large illumination angle and difference between the refractive indexes of cornea (1.376) and air (1.0), (ii) the shift of the coherence plane in the opposite direction towards the objectives, also caused by the refractive index difference (Fig. 3). The optical mismatch can be measured from the axial location of the cornea, provided in real-time by SDOCT. The mismatch is used to calculate, how far the reference arm of the interferometer should be extended (Fig. 2), in accordance with the defocus correction procedure[18], in order to match the coherence plane with a new focal position. This extension should be precise enough to land within the objective's depth of focus (measured ~ ±35 μm at FWHM) (Fig. 1b) and rapid enough to follow in vivo movements of the cornea, which is achieved using a voice coil motor (accuracy = 2.2 μm, velocity = 1 mm/s, acceleration ~25 mm/s$^2$) operating at a 50 Hz update rate. The entire combined FF/SD OCT system is mounted on a manual vertical and two motorized horizontal stages, controllable by the operator using a joystick.

**Imaging of ex vivo cornea, mimicking in vivo eye movements.** We tested and optimized real-time feedback of the common-path FF/SD OCT instrument by imaging ex vivo cornea (Fig. 4a), mounted on a moving stage. At first, the stage was programmed to produce a slow steady movement (20 μm/s) to compare system performance with enabled and disabled defocus correction. With defocus correction, as the SDOCT detects the cornea coming closer to the objective, and the image plane going deeper into the sample, the voice coil stage extends the reference arm to put it in

the optimal position for the current corneal location (Supplementary Movie 4). As a result, the FFOCT interferometer arms were well matched (Fig. 4f) with 3.7 ± 0.8 μm (mean ± s.d.) error well within the depth of focus, and FFOCT consistently displayed corneal images from various depths (Fig. 4g). Only occasionally the signal vanished due to phase changes induced by unwanted mechanical vibrations, caused by external factors (e.g., an underground metro passing nearby). Conversely, without defocus correction, the reference arm position remains fixed and FFOCT images are visible only at a single corneal position within the objective's depth of focus, which corresponds to matched optical path lengths of the interferometer arms (Fig. 4d, e and Supplementary Movie 4).

Next, we programmed the stage to move similarly to the physiological movements of the eye. This was achieved by first measuring the axial movements of the normal human cornea in vivo (Fig. 4b) and extracting the underlying frequencies and amplitudes (Fig. 4c). Two typical movements were visible: (i) heartbeat at 1.1 Hz with higher harmonics at 2.2, 3.3 Hz, etc., and (ii) slow breathing at 0.34 Hz, both in agreement with the literature[19]. We used the frequencies with the two highest amplitudes (0.34 Hz with 33 μm amplitude and 1.1 Hz with 13 μm amplitude) as our input for the motorized stage with the sample (Supplementary Movie 5). In the two cases, applying defocus correction led to positioning errors of only 1.5 ± 2.2 μm (mean ± s.d.) (Fig. 4h) and 5.3 ± 4.0 μm (mean ± s.d.) (Fig. 4i), i.e., within the depth of focus. Higher frequencies were not important, as the corresponding amplitudes were smaller than the axial resolution as well as the depth of focus of the FFOCT device.

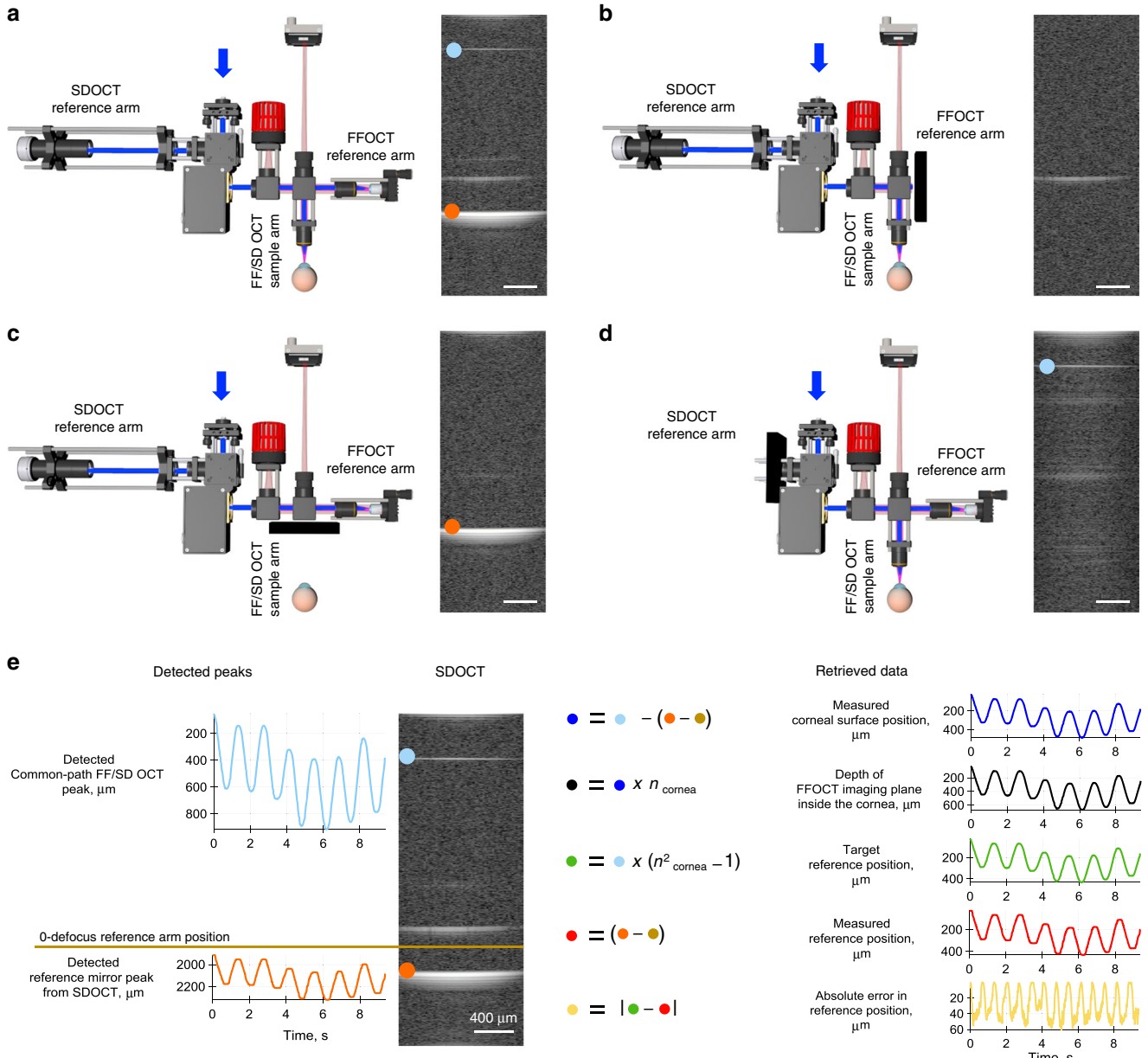

**Fig. 2 Real-time detection of back-scattering intensity in SDOCT and calculation of corneal and reference mirror locations. a** Left, common-path FF/SD OCT interferometer with light passing through all optical arms. **a** Right, SDOCT image with three planes of backscattered/reflected light. **b** Left, the same device with blocked FFOCT reference arm. Right, SDOCT image with a single reflected peak, corresponding to the surface of the cornea. **c** Left, device with blocked sample arm. Right, SDOCT image with reflection from the FFOCT reference mirror (orange dot). **d** Left, device with blocked SDOCT reference arm. Right, SDOCT image shows on top a single sharp reflection, originating from the FFOCT interference and captured by SDOCT (common-path peak, light-blue dot). Its narrowness reflects the perfect dispersion matching in identical arms of the FFOCT interferometer. **e** Left, SDOCT image with common-path and reference mirror peaks being detected in real-time (Supplementary Movie 3). Both of these maxima move down when the reference arm is extended for defocus correction. This facilitates their detection, as we can separate only the common-path peak for the upper area of the SDOCT image and the reference arm peak for the lower part without their overlapping. Right, using current locations of the two maxima and initial location of the reference mirror, one can calculate the corneal surface position in real-time, along with the depth of the FFOCT imaging plane in the cornea, the optimal reference position for the current corneal location, the actual reference position and the error between the two, used for validating and improving the optical arms matching loop. All scale bars are 400 μm. The source data underlying plots in **e** are provided as a Source Data file.

**Imaging of in vivo human cornea and sclera**. We used common-path FF/SD OCT to view in vivo human cornea in real-time. The study was carried out on three healthy subjects (1 female and 2 males, aged 36, 24, and 26 years respectively), which was confirmed by routine eye examination in the hospital preceding the experiment. Subjects expressed informed consent and the experimental procedures adhered to the tenets of the Declaration of Helsinki. Approval for the study was obtained, in conformity with French regulations, from CPP (Comité de Protection de Personnes) Sud-Est III de Bron and ANSM (Agence Nationale de Sécurité du Médicament et des Produits de Santé) study number 2019-A00942-55. During the experiment, subjects were asked to rest their chin and temples on a standard headrest, while looking at a fixation target. Examination was non-contact and without prior introduction of cycloplegic or mydriatic agents, nor topical anesthetics. While the system was heavily

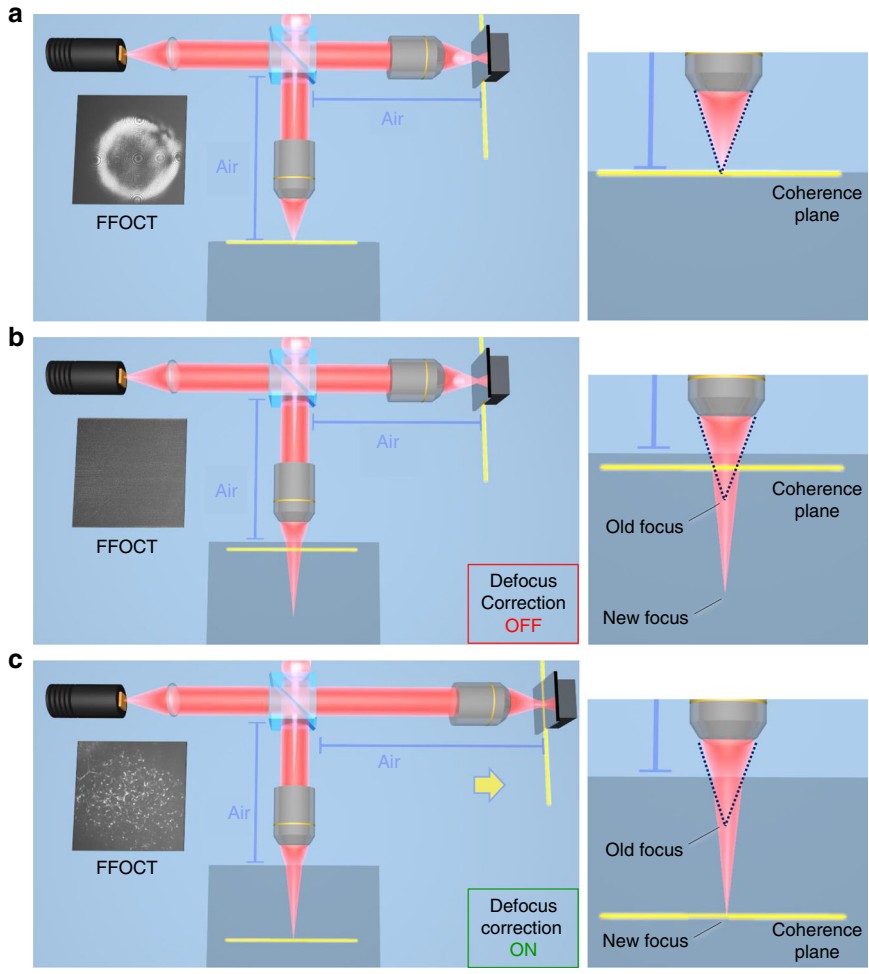

**Fig. 3 Principle of defocus correction for matching the optical arms of FFOCT interferometer. a** The FFOCT interferometer is focused on the surface of the sample (blue square). The focus spot matches with the coherence plane. An FFOCT image from the surface is captured and defocus correction is not needed. **b** FFOCT interferometer is focused inside the sample. Due to Snell's law the focus is extended. At the same time the coherence plane is shifted in the opposite direction due to the higher refractive index inside the sample than in the air. The focus spot and the coherence plane are mismatched and an FFOCT image cannot be captured without defocus correction. **c** The FFOCT interferometer is focused inside the sample and the focus is extended. The defocus correction procedure is performed: the reference arm is extended to put the coherence plane in the new location of the focus and the FFOCT image from inside of the sample is captured.

optimized for both fast acquisition rate and simultaneous display at 275 FFOCT images/s, we artificially lowered both acquisition and display rates of real-time imaging to 10 fps (with each frame captured in 3.5 ms) in order to be compliant with ocular safety standards. The pulsed light irradiance was below the maximum permissible exposure (MPE) levels of up-to-date ISO 15004-2:2007 (52% of MPE for cornea and 0.5% of MPE for retina) and ANSI Z80.36-2016 (3.7% of MPE for cornea and 0.5% of MPE for retina) (for more details see Methods section). Illumination was comfortable for viewing, due to the low sensitivity of the retina to NIR and IR light. The operator of the instrument could simultaneously view FFOCT movies of en face images, OCT movies with cross-sectional images of the main corneal reflex, which serves as an indicator of the current imaging depth, and plots illustrating the performance of the defocus correction. The instrument was able to acquire movies (Supplementary Movies 6, 7, 8, 9) from central, peripheral, and limbal zones of the in vivo cornea (Figs. 5, 6), simultaneously correcting defocus with $9.4 \pm 6.2\,\mu m$, $11.3 \pm 7.2\,\mu m$, and $7.2 \pm 6.6\,\mu m$ (mean ± s.d.) errors (Figs. 5e, 6b, c), respectively.

Non-averaged single FFOCT frames, extracted from the movies, had high signal from all corneal layers. Over the central

cornea we observed tear film. It appears with a fringe pattern, as FFOCT is an interferometric technique, meaning that interference fringes are visible on flat surfaces such as tear film (Fig. 5f). Just beneath the tear film, we could see superficial epithelial cells 40–50 μm in diameter with dark 8–13 μm nuclei, in agreement with the literature[20] (Fig. 5b, g). These cells were revealed by filtering the image in the Fourier domain (see Methods section). We could also see structures from other epithelial layers (wing, basal), however were unable to reliably resolve individual cells, due to low contrast between them. Beneath the epithelium, we saw the sub-basal nerve plexus (SNP) with 2–4 μm thick nerves in a vertical orientation (Fig. 5h), which is characteristic of the central corneal area, located superior to the whorl-like nerve pattern[21]. Due to the curvature of the cornea and small thickness of the SNP, only part of the layer was visible. Nevertheless, the area of the visible nerve section was 0.317 mm², three times larger compared to what is possible with the state-of-the-art IVCM. As a result, a large-scale view of SNP can potentially be obtained with a smaller number of "stitched" images[22], leading to significantly faster screening and processing times. We also show that clinically valuable[23] measurement of nerve density can be performed with FFOCT. Nerves were

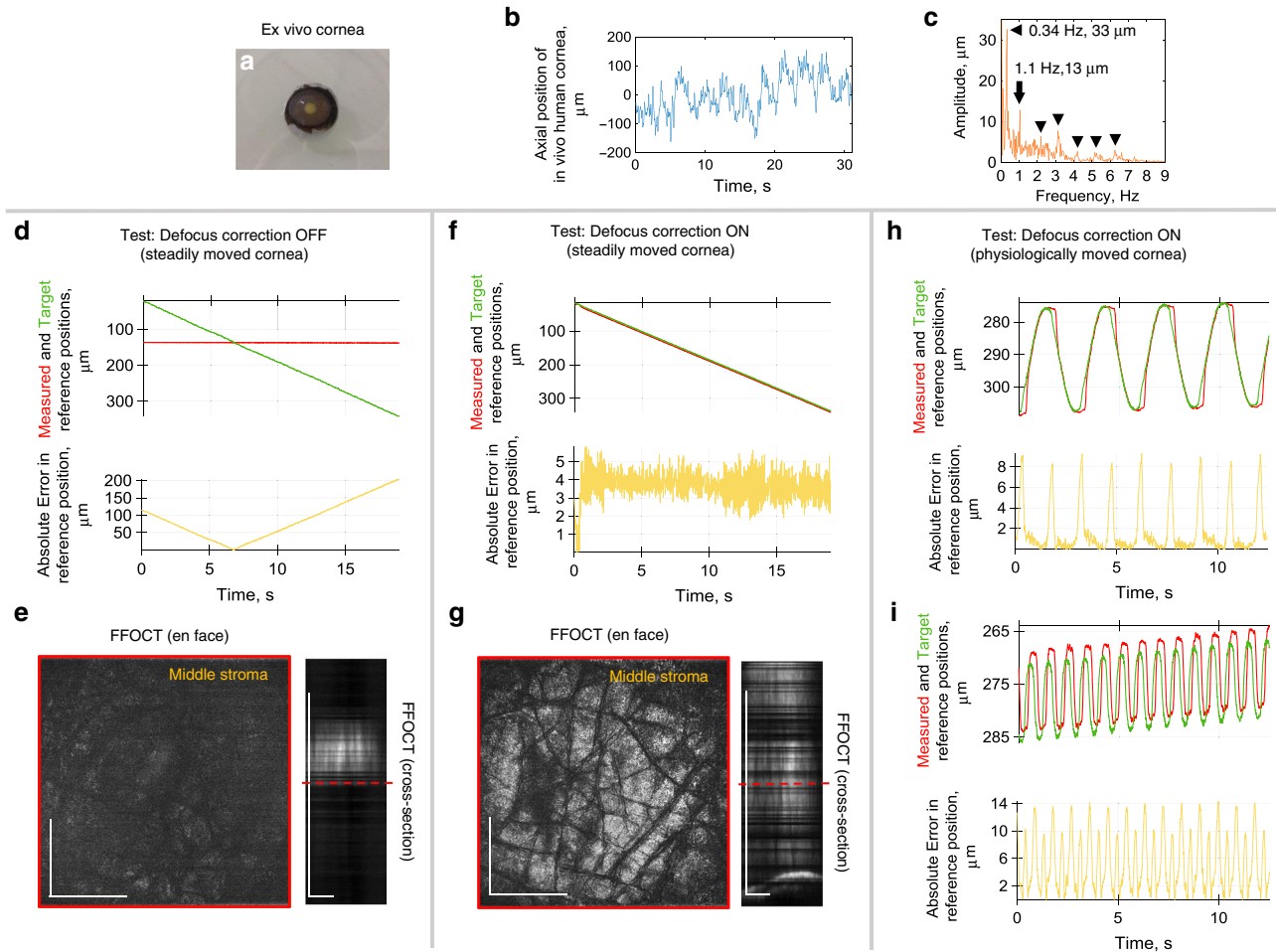

**Fig. 4 Validating the real-time defocusing correction on an ex vivo cornea, driven by the motorized stage. a** Photo of the corneal sample (without motor mount). Sample had visible stromal striae, indicative of tissue stress[49]. **b** Axial position of the in vivo human cornea over time, measured with SDOCT. **c** Extracted amplitudes and frequencies of the in vivo human corneal axial movements. **d** Steadily moved ex vivo cornea with deactivated defocus correction. Plots with measured reference mirror position (red), required reference location for ideal defocus correction (green) and error between the two (yellow). **e** Corresponding en face FFOCT and (reconstructed) cross-sectional FFOCT images. Red line shows location of en face FFOCT image. Without defocus correction, the reference arm position was fixed and FFOCT images were visible only at a single corneal position within the objective's depth of focus, where the optical path lengths of the interferometer arms match. **f** Steadily moved ex vivo cornea with active defocus correction. Small positioning error of 3.7 ± 0.8 μm (mean ± s.d.) was estimated from 2104 positions acquired over 19 s. **g** FFOCT images are consistently acquired from various depths, while only occasionally the signal vanishes due to the additional phase introduced to the tomographic signal by unwanted mechanical vibrations, caused by external factors (e.g., an underground metro passing nearby) (Supplementary Movie 4). **h** Test of defocusing correction with ex vivo cornea mimicking the physiological ocular displacement at 0.34 Hz and 33 μm amplitude. Positioning error of 1.5 ± 2.2 μm (mean ± s.d.), estimated from 1157 positions acquired over 12.4 s. **i** Test of defocusing correction with ex vivo cornea, moved at physiological 1.1 Hz and with 13 μm amplitude (Supplementary Movie 5). Positioning error is 5.3 ± 4.0 μm (mean ± s.d.), estimated from 1502 positions acquired over 12.5 s. The source data underlying plots in **b–d**, **f**, **h**, **i** are provided as a Source Data file. Experiments were reproduced several times using the same ex vivo cornea. All scale bars are 400 μm.

segmented using NeuronJ[24] and their density measured 15 mm/mm$^2$ (Fig. 5c), within the healthy margins[25]. Underneath the SNP, we enter the anterior part of the stroma with numerous bright oval-shaped nuclei of keratocyte cells, measured about 15 μm in diameter (Fig. 5i). From the anterior to posterior stroma the density of keratocytes gradually decreases down to the deep stromal layer, adjacent to Descemet's membrane, where the keratocyte density shows a slight increase, in agreement with the literature[26]. Moreover, with the increasing depth, the nuclei show a more elongated shape[27] (Fig. 5k, l). In addition to keratocyte nuclei, we can resolve keratocyte cell bodies and branching 10 μm thick stromal nerves (Fig. 5j, k). Descemet's membrane was also visible as a dark band separating stromal keratocytes from the endothelial cells (Fig. 5l). Central endothelium viewed in a single FFOCT image was hindered by a strong specular reflection and interference fringe artifacts; nevertheless, after lateral image registration and averaging (see Methods section) we could resolve the hexagonal mosaic of 20 μm diameter cells and sometimes a 5 μm nucleus (Fig. 5d, m). While the reflectivity of the endothelium-aqueous interface is expected to be 100 times less than the air-tear film interface (based on the refractive index difference), FFOCT images show only 10 times difference in brightness, because FFOCT acquires the images of amplitude (due to the 2-phase amplitude retrieval process) and not intensity like the conventional OCT. It should be noted that in the figures these interfaces appear with similar brightness due to an adjustment of brightness and contrast. We were able to perform clinically significant[28] cell counting and measured the normal endothelial cell density of 3096 cells/mm$^2$ (Fig. 5d), in agreement with the literature[29], and confirmed on the same subject using a clinical specular microscope, which counted 3100 cells/mm$^2$. All of the above images were acquired without any physical contact

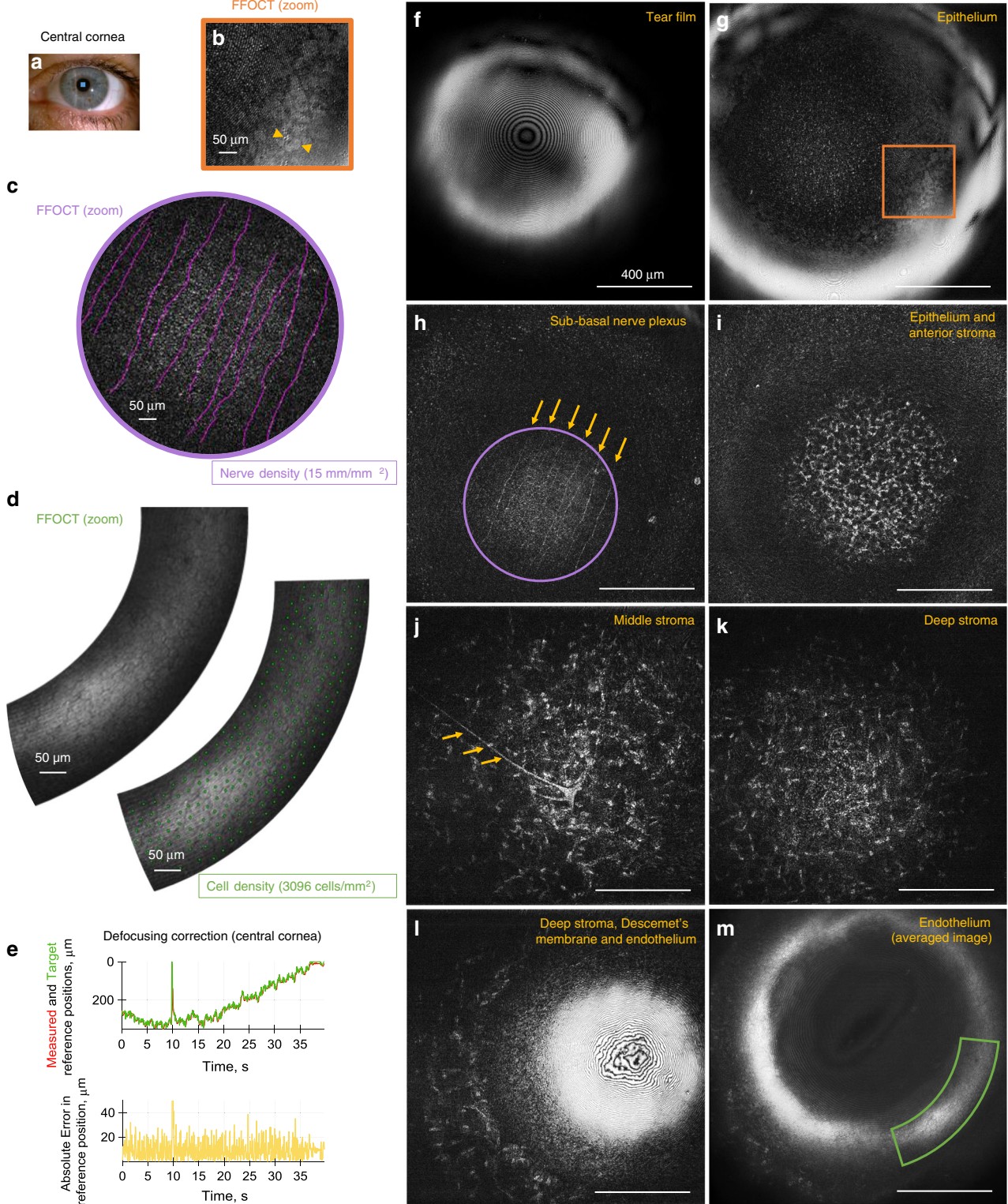

**Fig. 5 Common-path FF/SD OCT imaging of central human cornea in vivo. a** Slit-lamp macro photo obtained from one of the subjects. Blue square depicts the FFOCT field of view. **b** Zoomed and bandpass filtered (see Methods section) FFOCT image of superficial epithelial cells with dark nuclei. **c** Zoomed FFOCT image of sub-basal nerves. NeuronJ[24] was used for nerve tracing and quantification. **d** Zoomed FFOCT image of endothelial cell mosaic. ImageJ[47] point tool was used for cell counting. **e** Performance of real-time defocus correction, when imaging in vivo human corneal center. Measured reference mirror position (red), required reference location for ideal defocus correction (green) and error between the two (yellow). The peak at 10 s corresponds to a blink. Positioning error was 9.4 ± 6.2 μm (mean ± s.d.), estimated from 1488 positions acquired over 38.6 s. **f–m** Single frame FFOCT images through the entire thickness of the central cornea extracted from the real-time movies (Supplementary Movies 6, 7). Surface epithelial cells, sub-basal, and stromal nerves (yellow arrows), keratocyte cells and their nuclei, and endothelial cells were visible. The source data underlying plots in **e** are provided as a Source Data file. All unlabeled scale bars 400 μm. Experiments were reproduced several times in each of the three healthy subjects.

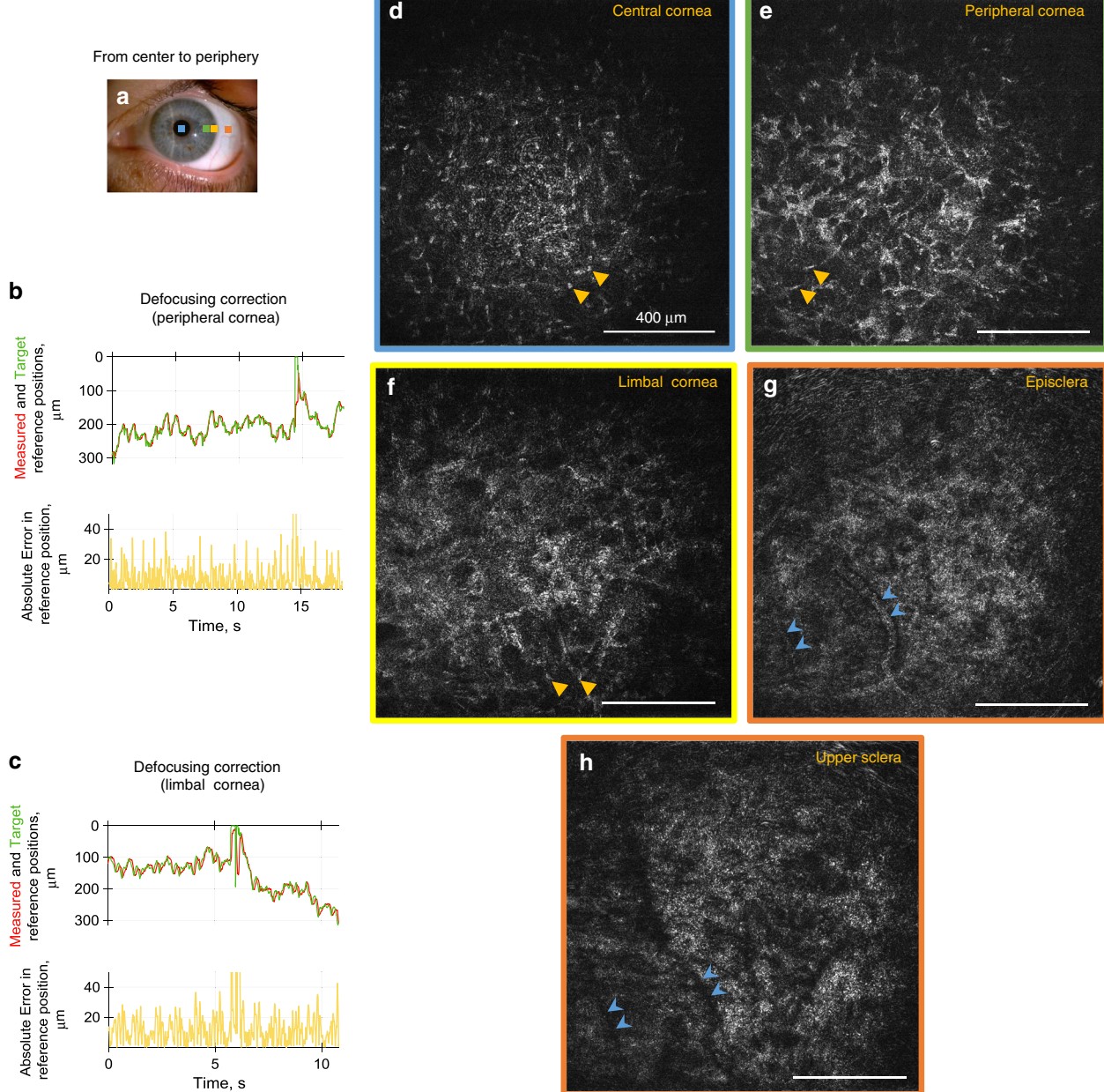

**Fig. 6 Common-path FF/SD OCT imaging of peripheral human cornea, episclera and upper sclera in vivo. a** Slit-lamp macro photo obtained from one of the subjects. Squares depict the peripheral locations, where FFOCT images were acquired. **b**, **c** Performance of real-time defocus correction, when imaging the corneal periphery. Measured reference mirror position (red), required reference location for ideal defocus correction (green) and error between the two (yellow). Peaks at 6 and 14 s correspond to blinks. Positioning errors were 11.3 ± 7.2 μm and 7.2 ± 6.6 μm (mean ± s.d.) estimated from 1022 and 1222 positions acquired over 18.2 and 10.8 s, respectively. **d–h** Single frame FFOCT images from central cornea to periphery, extracted from real-time movies (Supplementary Movies 8, 9). Resolving individual keratocyte nuclei (yellow arrows) was increasingly more difficult, when imaging further from the center, due to stronger light scattering from the stromal fibrils and consequent glare over the entire image. Blue arrows show vessels of episclera, which produced shadows in the upper sclera. The source data underlying plots in **b** and **c** are provided as a Source Data file. All unlabeled scale bars 400 μm. Experiments were reproduced several times in each of the three healthy subjects.

with the eye, with a distance of about 2 cm between the cornea and the microscope objective. We also benefited from the insensitivity of FFOCT to aberrations[30]. More precisely, the spatially incoherent light source in full-field illumination ensures that the non-aberrated light from the reference arm can effectively interfere only with a non-aberrated portion of light from the sample, while interference with the aberrated part is heavily suppressed[31]. As a result, FFOCT keeps the diffraction-limited 1.7 μm resolution through the entire cornea, despite the presence of spherical aberrations, which are expected to reduce

the resolution of conventional OCT and IVCM systems by three times to 5.1 μm (at the paraxial focus 550 μm deep inside the cornea). Nonetheless, it should be noted that both spectrally-dependent scattering and absorption in biological tissue, that is also spatially-dependent, do affect the FFOCT axial resolution.

We also looked at the appearance of stroma in the central and peripheral cornea, and sclera. The dark background of the central corneal stroma (Fig. 6d), becomes bright at the periphery (Fig. 6e, f), which is explained by the increased light scattering from the stromal fibrils, irregular in diameter and arrangement, known

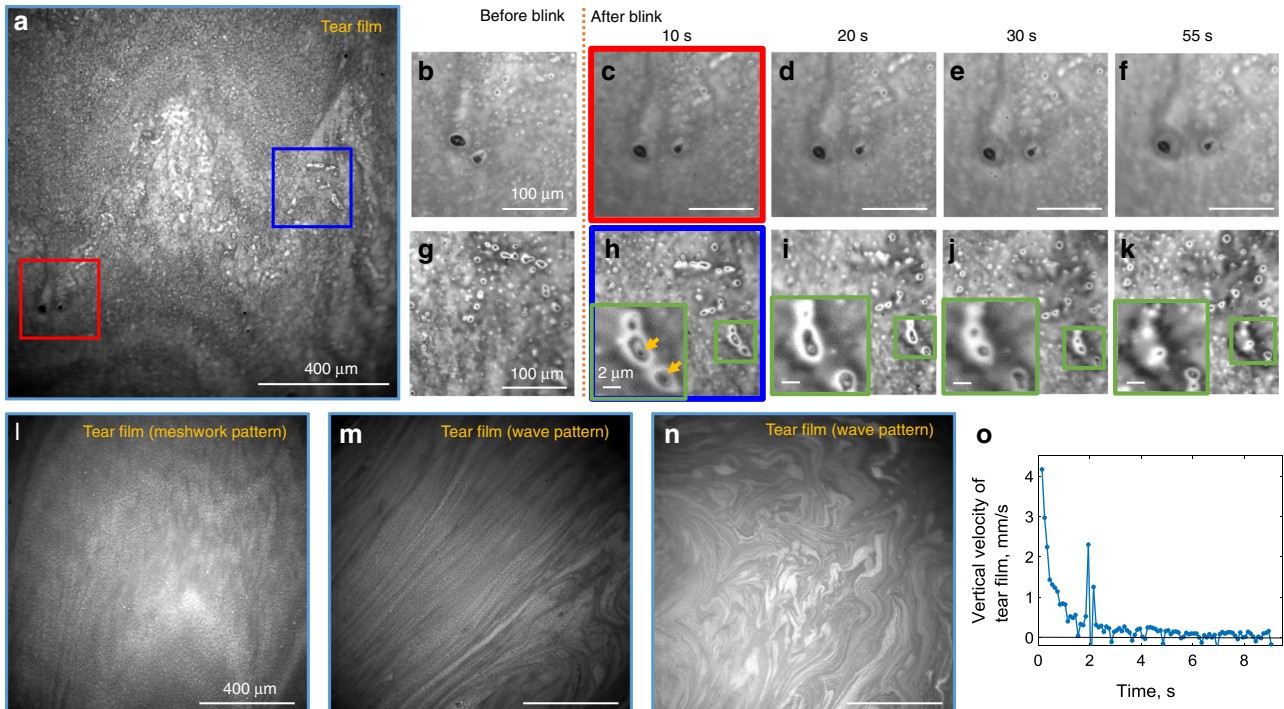

**Fig. 7 Imaging human tear film in vivo with conventional microscope configuration of FF/SD OCT. a, l–n** Single frame direct reflection images from the tear film. Interference patterns form the tear lipid layer can be used to evaluate its thickness. **b–f**, **g–k** Zoomed images from the real-time blink movie (Supplementary Movie 10). Isolated particulate matter, thought to be cellular debris or accumulations of newly secreted lipid from the Meibomian glands, were found in all subjects. Particles were static, changing locations only from blink to blink together with the movement of the tear film. Green zoomed images show liquid drops surrounding small particles. Liquid was evaporating over time. **o** Vertical velocity of the tear film, measured by manually tracking movements of particles in the movie. Tear film stabilized after 9 s following the blink. Peak at 2 s corresponds to saccadic eye motion. The source data underlying the plot in **o** are provided as a Source Data file. All unlabeled scale bars 400 μm. Experiments were reproduced several times in each of the three healthy subjects.

from electron microscopy studies[32]. The aforementioned scattering bright glare of the background makes it more difficult to resolve keratocyte cell nuclei at the periphery, comparing to those in the central cornea, easily visible against the dark background, in agreement with previous confocal microscopy data[33]. Blood vessels (Fig. 6g) were perforating the conjunctiva and episclera and produced shadows (Fig. 6h) in the upper sections of the sclera.

It should be noted that ocular movements not only shift the position of the eye leading to the problem of defocus, but they also intervene in the FFOCT image retrieval scheme, affecting image quality. The rapid axial motion of the eye can introduce additional phase modulation, which, when added to the π piezo modulation, reduces the FFOCT signal. This effect is seen in FFOCT movies as occasional wash out of the tomographic signal (Supplementary Movies 6–9). The rapid saccadic lateral motion of the eye as well as the rapid flow of tears after a blink can also introduce the artifacts to the FFOCT images. More precisely, the two-phase retrieval scheme is unable to completely remove the light originating from outside of the coherence volume, because the scene in the two images is shifted. As the largest proportion of out-of-coherence-volume light comes from the air-tear film interface, the artifacts manifest the defocused view of the ocular surface. The artifacts occasionally appear in the FFOCT movies (Supplementary Movies 6–9) and are more seldom present when imaging the deeper corneal layers (deep stroma, endothelium), as the surface gets further from the focus of the optical system.

**Imaging of in vivo human tear film.** We were able to demonstrate tear film evolution by blocking the reference arm, thus

converting our FF/SD OCT setup into a conventional microscope. In order to increase contrast, an image of stray light from the beam splitter, acquired without the sample, was subtracted from the tear film image. Before each examination, subjects were asked to keep their eyes closed for 2 min to replenish the tear film. Right after the eye opened, a wave pattern was typically observed (Fig. 7m, n), originating from the interference within the tear film lipid layer[34]. In the normal condition (opened eyes, free to blink) we often saw meshwork (Fig. 7a, l) patterns. We acquired movies of tear flow after a blink (Supplementary Movie 10) and after a half-blink, when the eye was not completely closed, but the tear film shifted (Supplementary Movie 11). With a blink, the upper lid rapidly moved upward and the layer of tears followed with a slight delay. At 150 ms after the blink, the flow velocity was 4.2 mm/s and rapidly decreased to 0.8 mm/s after 1 s, completely stabilizing to zero in 9 s, in agreement with literature[35,36] (Fig. 7o). Isolated particulate matter, about 1–40 μm in size, was found in all the subjects in lipid and aqueous layers (Fig. 7a, b, g), thought to be cellular debris or accumulations of newly secreted lipid from the Meibomian glands[35]. Particles were static (Fig. 7c–f, h–k), changing locations only from blink to blink together with the movement of the tear film (Fig. 7b, c, g, h). We also noticed that small particles were frequently surrounded by liquid drops, which evaporated over time (Fig. 7h–k).

**Imaging of in vivo human limbus.** In the inferior limbal region, 30 μm wide radial palisades of Vogt (POV) were visible (Fig. 8f). The distance between the palisades measured 30–200 μm. Marginal corneal vascular arcades (MCA) with thin 3–7 μm vessels

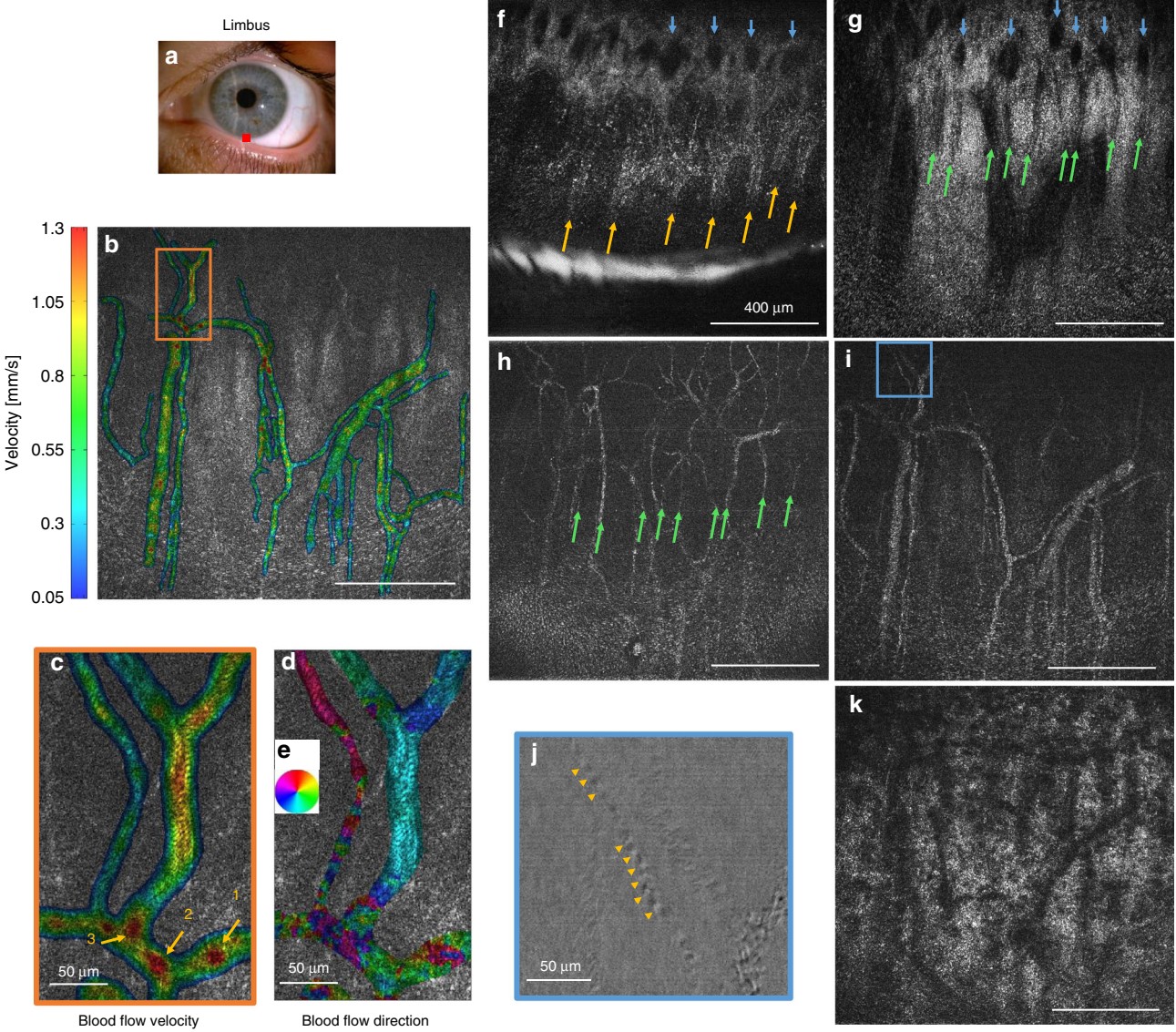

**Fig. 8 Common-path FF/SD OCT imaging and angiography of in vivo human limbus. a** Slit-lamp macro photo obtained from one of the subjects. Red square depicts location, where FFOCT images were acquired. **b, c** Blood flow velocity map, retrieved from the rapidly acquired movie at 275 images/s (Supplementary Movie 12), together with a zoomed image. The lowest speeds were measured close to vessel walls and highest speeds in the middle of the vessels, in junction points (point 2 and point 1 (with merging vessel coming to point 1 visible only in the movie)) and also, as artifacts, if vessels overlaid each other (point 3). **d** Blood flow orientation map retrieved from the movie. Each color corresponds to a certain direction of blood flow, according to the colormap **e**. **f–k** Single frame FFOCT images of consecutive depths in limbus. Yellow arrows—palisades, green arrows—thin vessels hosted within palisades, blue arrows—perpendicular vessel network thought to be connected with horizontal vessels. Underneath, thicker vessels and scattering stroma-sclera medium with vessel shadows were seen. **j** Zoomed view on thin vessel in **i**. Image is obtained by subtracting the two camera shots. Contrast is reduced (see Methods), but is more intuitive for resolving and following individual blood cells (yellow arrows) (Supplementary Movie 12). Unlabeled scale bars are 400 μm. Experiments were reproduced several times in each of the three healthy subjects.

(Fig. 8g, h) appeared to protrude from inside of the palisades, in agreement with the literature[16,37,38]. These vessels, parallel to the ocular surface, appeared to be connected with a perpendicular oriented vessel network, visible as dark round shadows (Fig. 8f, g). Closer to the cornea, MCA vessels curled into the loops, while continuing to spread in the radial direction. Beneath, we could see thicker 40 μm branching vessels (Fig. 8i). By looking at the difference between two camera images (see Methods section), it was possible to resolve individual 7 μm blood cells (Fig. 8j and Supplementary Movie 12), which were difficult to see with a tomographic FFOCT image. Moreover, using rapid 275 frame/s acquisition, we could visualize and track the flow of these cells,

which had a speed of about 1 mm/s (Supplementary Movie 12). Furthermore, in order to capture the full-field information about the blood flow, important for addressing inflammatory conditions in the anterior eye, we measured the local cross-correlation for each 16 pixel × 16 pixel sub-image and retrieved blood flow velocity and blood flow direction maps with micrometer-level resolution (Fig. 8b–e). Average velocity was measured to be 0.446 ± 0.270 mm/s, with the lowest speeds close to vessel walls and highest speeds in the middle of the vessels, at junctions and also in locations where vessels were overlaying each other (Fig. 8c). Blood flow was mostly radial, going back and forth from the cornea (Fig. 8d, e).

## Discussion

Confocal microscopy, the state-of-the-art tool for cellular-resolution imaging of the cornea, requires contact with the eye, preceded by ocular anesthesia. Conventional OCT provides high axial resolution of corneal layers, but does not resolve cells, while UHR-OCT resolves cells, however the en face images suffer from eye movement artifacts during X–Y beam scanning. Our common-path FF/SD OCT reveals the same micrometer corneal features (cell, nerves, and nuclei), but in a completely non-contact way, with a 2 cm distance from the eye, does not require use of any medication and is immune to beam scanning artifacts. These features improve patient comfort, remove risk of corneal damage and risk of infection, and open up the possibility for high-resolution imaging in a risk-sensitive population (e.g., young children, candidates for corneal transplant surgery with fragile corneas). Compared to our previous preliminary work[10], in this article we have integrated SDOCT into FFOCT, which allowed tracking of the eye position and matching the arms of the FFOCT interferometer in real-time. This has enabled consistent real-time acquisition of corneal images and movies with high signal and allowed exploration of the entire ocular surface (central, peripheral, and limbal cornea, as well as limbus, episclera and sclera). This opens a path for FFOCT implementation in clinical research and translation into clinical practice. Our device provides images with three times larger field of view than IVCM (nine times larger in area), which makes it easier to locate the clinical area of interest and follow the same location over time. Furthermore, a larger viewing area is useful for quantitative measurements of sub-basal nerve and endothelial cell densities, which were demonstrated with common-path FF/SD OCT following methodologies, used in confocal studies. Both are clinically valuable parameters: nerve density correlates with keratoconus, dry eye, several types of keratitis and diabetes[39], while endothelial cell density correlates with dystrophies, uveitis and acute ischemic stroke[40]. As FFOCT lateral resolution is immune to aberrations[31], the device is expected to resolve nerves and cells in pathological distorted corneas as well (e.g., in the case of keratoconus). Axial resolution could be improved (for resolving finer corneal layers) in future iterations of the device through increasing the spectral bandwidth of the source, for example by implementing emerging broadband NIR LEDs of sufficient power or by combining several existing NIR LEDs with close spectral characteristics.

The instrument also opens a door to quantitative diagnosis of dry eye condition by providing information about tear film velocity and stabilization time following a blink, the evaporation time of the liquid micro-droplets on the surface of the eye, and potentially the thickness of the tear film by grading lipid interference patterns in a similar way to the Guillon Keeler tear film grading system[41]. Common-path FF/SD OCT can resolve finer high-resolution sections of the limbus and is able to distinguish layers of palisades of Vogt and underlying vessels, compared to UHR-OCT[16] and IVCM[42]. Furthermore, our instrument is not limited to diagnosis of "static" corneal disorders, but can potentially monitor inflammatory and scarring conditions affecting the fast (millisecond) dynamics of the blood flow in the eye. Previously, general information about the blood flow velocity and vessel dimensions from anterior segment fluorescein angiography (ASFA) and indocyanine green dye angiography (IGDA) proved to be useful for distinguishing between the various forms of scleral inflammation, in particular between severe episcleritis and diffuse anterior scleritis or between peripheral corneal opacification and corneal thinning[43]. With FF/SD OCT, we can have access to the same information and view the blood flow with high contrast, but without fluorescein injection and with the examination taking a fraction of a second. Furthermore, the ability of the instrument to capture high-resolution maps of blood flow velocities and orientations can potentially be useful for opening up new localized diagnostic methodologies and ways to monitor effects of therapies locally. This could bring insights about the anatomy and physiology of the vascular system in the anterior eye and in particular the limbal vessels and MCA supplying the cornea. Moreover, we suggest that the intravascular mapping of blood flow with our device may be a promising platform for testing in vivo microfluidic theories[44]. Last, but not least, common-path FF/SD OCT can also be used for non-contact examination of various in vivo human, animal and ex vivo tissues.

## Methods

**FFOCT device.** The FFOCT device is based on an interference microscope in a Linnik configuration with identical microscope objectives in the two arms of the interferometer. Objectives (LMPLN10XIR, Olympus, Japan) have a numerical aperture (NA) of 0.3, 10× magnification and give high lateral resolution of 1.7 µm (with a filled 10.8 mm entrance aperture). The lateral resolution (identical in cornea and in free space) was estimated according to the Rayleigh criterion and experimentally confirmed with a resolution target and by measuring the diameters (FWHM) of 80 nm gold nanoparticles imaged on a glass plate, located at the focal plane of the FFOCT objective. The working distance of the objectives was 18 mm, sufficient to avoid the risk of accidental physical contact with the eye. Illumination is provided by an NIR 850 nm light-emitting diode (LED) source (M850LP1, Thorlabs, USA). The axial resolution of 7.7 µm in the cornea was estimated from the experimentally measured spectral bandwidth of the LED (30 nm) with a spectrometer (CCS175/M, Thorlabs, USA) and by using the average corneal refractive index of 1.376, often quoted in the literature[45]. Light from the LED is collected by an aspheric condenser lens (ACL12708U-B, Thorlabs, USA) and is focused on the back focal plane of the objective. Before entering the objective, light from the source is equally separated by the 50:50 beam splitter cube (BS) (BS014, Thorlabs, USA) into the sample and reference arms of the interferometer. The arms are slightly tilted from a perpendicular orientation in order to avoid specular reflection from the BS side. The objective in the reference arm focuses light onto an absorptive neutral density (ND) glass filter (NENIR550B, Thorlabs, USA), which plays the role of a single mirrored surface with 4% reflectivity. Use of an ND filter with OD 5.0 instead of a glass plate enables elimination of ghost reflections from the back surface of the filter. A low reflectivity value is chosen to achieve high detection sensitivity, which is maximized when the total reflectivity (from all the layers participating or not to the FFOCT signal) of the sample and reflectivity of the reference mirror match, as indicated by FFOCT signal to noise calculations[12]. Total reflectivity from all corneal layers, estimated from the Fresnel relations, is around 2%. By using a reference mirror with a reflectivity of 4%, we can expect sensitivity close to the ideal condition[12]. Light in the sample arm, backscattered from the different planes in the cornea, and light in the reference arm, backscattered from a single mirror plane, are collected by the objectives, and recombined on the BS. This results in interference, but only for the light coming from the corneal plane, and light coming from the reference mirror plane, which match in terms of optical path length. The temporal coherence length of the light source determines the thickness of the interference fringe axial extension and therefore the optical sectioning precision, which is 7.7 µm. Interfering light and non-interfering light, arising from other planes of the cornea, are focused onto a sensor by a magnifying tube lens (AC254-250-B, Thorlabs, USA), leading to 14× overall system magnification. The sensor is a high-full well capacity (2Me⁻) 1440 × 1440 pixel CMOS camera (Q-2A750-CXP, Adimec, Netherlands), which captures each 2D image in 1.75 ms. In order to reach the expected axial sectioning in the system, light which has interfered must be extracted from the background. To do so, we use a two-phase shifting scheme, where we rapidly modulate the position of the reference mirror using a piezo mirror-shifter (STr-25/150/6, Piezomechanik GmbH, Germany), synchronized with the camera, to capture two consecutive images, which have a π phase shift, and subtract them. The absolute value of the resulting image contains only interfering light from a 7.7 µm thick section in the cornea. We can capture images in two modes: (i) fast acquisition mode for visualizing blood flow, where a sequence of 40 images is captured at 550 fps, and (ii) slow mode for real-time imaging, where images are displayed live on screen at 10 fps. The real-time mode has artificially lowered acquisition and display rate to 10 fps in order to be compliant with the European ISO 15004-2:2007 ocular safety standard (see below). 10 fast pulsed LED exposures/s (3.5 ms each exposure) with pauses in between each exposure allow us to increase the output of the LED to saturate the high full-well capacity camera, while still being below the maximum permissible light levels. It should be noted that the most recent US ocular safety standard ANSI Z80.36-2016 imposes a much less strict limitation for exposure of the anterior eye, which in the future potentially enables the prolonged use of our device at full speed frame rate at 275 fps without LED pulsation.

A complete evaluation of corneal and retinal safety, involving scenarios of real-time imaging (used to visualize static corneal structures) and single prolonged LED exposure (used to visualize blood flow dynamics), was performed. The results are summarized below.

For real-time imaging we took into consideration the durations of pulsed light exposures (3.5 ms for LED and 1.37 ms for SLD (beam scanning)), durations of pauses without the light (100 ms for LED and 0.41 ms for SLD) and irradiances per pulse (2 W/cm² for LED and 35 mW/cm² for SLD). These values were used to calculate the corneal and retinal exposures. The calculated time-dependent graphs show that during real-time acquisition we reach up to 52% of MPE for the cornea and only 0.5% of MPE for the retina (according to ISO 15004-2:2007) and 3.7% of MPE for the cornea and 0.5% of MPE for the retina (ANSI Z80.36-2016).

For single prolonged LED exposure we took into consideration the durations of pulsed light exposures (70 ms for LED and 1.37 ms for SLD (beam scanning)), durations of pauses without the light (30 s for LED and 0.41 ms for SLD) and irradiances per pulse (2 W/cm² for LED and 35 mW/cm² for SLD). The calculated time-dependent graphs show that during real-time acquisition we reach up to 18% of MPE for the cornea and only 1% of MPE for the retina (according to ISO 15004-2:2007) and 1.3% of MPE for the cornea and 1% of MPE for the retina (ANSI Z80.36-2016). The major contributor to the exposure is the LED.

**SDOCT device.** The SDOCT device is based on a commercial general-purpose spectral-domain OCT (GAN510, Thorlabs, USA). It consists of an interferometer with a galvanometric mirror system (OCTP-900(/M), Thorlabs, USA), which rapidly scans a collimated light beam laterally at 100 kHz to form a 2D cross-sectional image. In order to increase the frame-rate, the lateral extension of the image was limited to 64 pixels. The 2D image is averaged over the lateral dimension and the resulting 1D data is processed with Labview Peak detector VI (National Instruments, USA) to locate the maxima. As these are single non-overlapping peaks, their positions can be detected with a precision better than 3.9 μm in the cornea, as determined by the Rayleigh criterion and spectral bandwidth of infrared (IR) 930 nm superluminescent diode (SLD). Because the SDOCT and FFOCT interferometers share the same optical arms, we detect not only the conventional SDOCT peaks corresponding to the reflection from the cornea and from the FFOCT reference mirror, but also the third common-path peak corresponding to the interference inside the FFOCT interferometer (i.e., between FFOCT sample and FFOCT reference arms). The latter maximum has the advantages that (i) it does not suffer from dispersion, due to the perfect dispersion symmetry between the arms of the FFOCT interferometer; (ii) it is moving in the same direction on the 2D image as the reference arm, extended for defocus correction. As a result, since the common-path and reference mirror peaks never overlap, we can simultaneously detect them and calculate positions of the cornea, the imaging depth inside the cornea, the optimal reference position for the defocus correction[18], the actual reference position and the error for validating and improving the feedback loop. By reducing the light entering into the reference arm of SDOCT, we suppress the conventional peak from the cornea to further facilitate acquisition of common-path and conventional reference arm peaks. Maxima were detected over a 1.25 mm lateral range, matching the FFOCT field of view and over a 2.7 mm axial range, determined by the SDOCT spectrometer. Information about the locations of the peaks was refreshed every 8.2 ± 0.5 ms (mean ± standard deviation (s.d.)).

**Integrated FFOCT-SDOCT instrument.** SDOCT is optically integrated through the dichroic mirror (Edmund Optics, USA) into the illumination arm of FFOCT. In order to block the SDOCT light from reaching the FFOCT camera, two filters with opposite spectral characteristics (low and high pass at 900 nm cutoff) (FELH0900 and FESH0900, Thorlabs, USA) are positioned at the entrance and exit of the FFOCT device. Glass windows (WG11050-B, Thorlabs, USA) are inserted into the reference arm of SDOCT to dispersion match with the FFOCT sample and reference arms. The combined instrument is positioned on two high-load lateral translation stages (NRT150/M, Thorlabs, USA), controlled by a driver (BSC202, Thorlabs, USA) with a joystick (MJC001, Thorlabs, USA). Beneath, a laboratory jack (L490/M, Thorlabs, USA) is used to position the whole device vertically. The reference arm of the FFOCT device is mounted on a voice-coil translation stage (X-DMQ12P-DE52, Zaber, Canada), driven with 2.2 μm accuracy, 1 mm/s velocity and 25 mm/s² acceleration for rapid defocus adjustment. Information about the currently required defocus correction shift is communicated to the stage from the SDOCT system through the same personal computer (PC) to avoid time delays. The weak link in the communication is the limited read frequency of the stage encoder, which can accept 50 new positions/s—about two times slower compared to the rate of positions provided by SDOCT. SDOCT image acquisition is controlled by a separate PC, which is synchronized with the FFOCT PC using the 11 ± 3 ms (mean ± s.d.) precision NI-PSP protocol via the local network of the Langevin Institute. Generation of 3D images should be possible in the future generations of the device with FFOCT and SDOCT being driven by a single PC.

**Software.** Custom programs written in Labview 2014 and Thorlabs SpectralRadar SDK were used for FFOCT and SDOCT image acquisition and display, peak detection and motor control. MATLAB R2017a was used for plots and calculation of angiography maps. We utilized ImageJ 1.51p for image display and measurements of cell and nerve densities. ZEMAX Optics Studio was used to simulate the light propagation inside the cornea and measure aberrations.

**Artifact suppression and contrast enhancement.** In order to remove surface interference fringes and reveal superficial epithelial cells (Fig. 5b), we transformed the image into the Fourier domain and masked the bright spots, corresponding to fringes. Inverting this Fourier image with suppressed artifacts revealed the cells. The same artifacts were also visible at the outer endothelial layer. We were able to suppress them by registering and averaging multiple (23) tomographic FFOCT images with ImageJ[46] without Fourier domain conversion (although averaging and Fourier domain processing can be used together to improve the final image). It is worth noting that filtering the interference fringes was more difficult in cases, when the endothelium was captured exactly in line with the corneal apex, because the fringes had varying spatial frequency and spacing. The endothelium on the apex had the smallest fringe frequency, because its orientation was perpendicular to the optical axis, i.e.,—the same orientation as the reference mirror. The endothelium elsewhere than at the apex was tilted due to the corneal curvature, which increased the interference fringe frequency.

The image and movie (Fig. 8j and Supplementary Movie 12) of individual blood cells was obtained not with a FFOCT image retrieval scheme, but by subtracting two consecutive images from the camera without taking the absolute value. The reason for this is the following: when we retrieve the FFOCT image via the two-phase method, the two consecutive images from the camera are subtracted and then the absolute of the image is taken to reveal the pixels, previously hidden due to their negative value after subtraction. Unfortunately, in the case of FFOCT imaging of a moving particle, we get a "doubling artifact", i.e. we see the moving particle as the two objects separated by a small gap. This complicates particle tracking. To get a better view of moving particles, we performed image subtraction without taking the absolute value. In this case each particle is seen as a single object, which simplifies its tracking. However, half of the signal (i.e., corresponding to the negative pixels) is lost.

**Quantitative image analysis.** Semi-automated nerve segmentation and density analysis (Fig. 5c) was performed with ImageJ[47] using the NeuronJ plugin[24]. Manual endothelial cell counting (Fig. 5d) was done with the Multi-point Tool in ImageJ. A Manual Tracking ImageJ plugin enabled manual blood cell tracking and the subsequent particle velocity analysis (Supplementary Movie 12).

**Mapping blood flow velocity and orientation.** Previously, we demonstrated the possibility of local blood flow measurements from the conjunctival surface using manually drawn kymograph plots (i.e., plotting the vessel curvilinear abscissa against time)[48]. However, each plot provided only a single local velocity value, so that obtaining a full-field velocity map would require considerable manual effort. Here we used a method based on a block-matching algorithm to track single features inside vessels, which allows rapid semi-automatic mapping of blood flow velocities and orientations in full-field. We cross-correlated 16 × 16 pixels windows and retrieved the cross-correlation maximum for each pixel in the vicinity corresponding to a maximal speed of 2 mm/s. Low cross-correlation peaks were discarded in order to remove artifacts and outliers. A velocity array is then created based on eight frames, so that the velocity computed for each pixel is the average of the seven velocities computed by the block-matching algorithm. Then the velocities and orientations are mapped on the hue channel, the FFOCT image is mapped on the value channel and the saturation is arbitrarily set to 0.8 for each pixel in order to construct the velocity and orientation map of the blood flow (Fig. 8b–e).

**Ex vivo cornea.** Ex vivo macaque corneas were obtained from a partner research institution as recuperated waste tissue from an unrelated experiment. Corneas were dissected from the ocular globes within two hours post-mortem and fixed in 2% paraformaldehyde prior to transfer to the imaging lab. Some edematous swelling occurred, causing enhanced visibility of stromal striae[49], indicative of tissue stress.

**In vivo imaging.** Informed consent was obtained from all subjects and the experimental procedures adhered to the tenets of the Declaration of Helsinki. Examination was non-contact and no medication was introduced into the eye. Light illumination, visible as a dim red circular background, was comfortable for viewing, due to the low sensitivity of the retina to NIR and IR light. For real-time imaging, light exposures measured 52% of maximum permissible exposure (MPE) for cornea and only 0.5% of MPE for retina (according to ISO 15004-2:2007) and 3.7% of MPE for the cornea and 0.5% of MPE for the retina (ANSI Z80.36-2016). For single exposure imaging (angiography), light exposures measured 18% of maximum permissible exposure (MPE) for cornea and only 1% of MPE for retina (according to ISO 15004-2:2007) and 1.3% of MPE for the cornea and 1% of MPE for the retina (ANSI Z80.36-2016). These values reflect that the light beam is focused on the cornea and widely spread on the retina. Up-to-date ISO and ANSI standards specify different MPE levels for corneal imaging at 850 nm wavelength, leading to different safety margins. The subject's head was comfortably positioned with temple supports and a chin rest. While one eye was imaged, the second eye was fixating on a target. When imaging non-central parts of the cornea, the subject's head was tilted by the examiner to position the eye's surface plane perpendicular to the direction of the incoming light beam.

**Other instruments**. Macro images were obtained with a slit-lamp biomicroscope (Topcon, France Medical S.A.S.) using 10× magnification and the lowest illumination. For comparison with FFOCT images, the endothelium was photographed using a clinical specular microscope (SP-3000P, Topcon, Japan) with 0.25 × 0.5 mm field of view.

**Reporting summary**. Further information on research design is available in the Nature Research Reporting Summary linked to this article.

## Data availability

The data that support the findings of this study are available from the first author and the corresponding author upon request. The source data underlying plots in Figs. 1b, 2e, 4b–d, f, h, i, 5e, 6b, c, and 7o are provided as a Source Data file.

## Code availability

Code that supports the findings of this study is available upon reasonable request from the first author but third party restrictions may apply (e.g., Thorlabs SDK).

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

## Acknowledgements

This work was supported by the HELMHOLTZ synergy grant funded by the European Research Council (ERC) (610110) and by a CNRS pre-maturation grant, as well as by the European Union's Horizon 2020 research and innovation program under the Marie Skłodowska-Curie grant agreement No 709104 (K.I.). We are grateful to Cristina Georgeon and Marie Borderie for assistance with acquiring the slit-lamp and specular microscope images. We thank Michel Paques and Jean-Marie Chassot for valuable discussions.

## Author contributions

V.M., M.F., and A.C.B. conceptualized the general idea. V.M., P.X., and A.C.B conceived and developed the optical design. V.M., J.S., and A.C.B. conceived the software and

hardware architecture. V.M. wrote the software and performed the imaging experiments. K.I. and K.G. provided the corneal samples and guidance on usage and experiments. V.M., K.I., K.G., and A.C.B. analyzed the acquired data. J.S. conceived and developed algorithms for obtaining quantitative blood flow velocity and orientation maps. K.G. (native English speaker) edited for language. V.M. wrote the manuscript, and all the authors contributed with edits and revisions.

## Competing interests

The authors declare no competing interests.
