## [Peer Review File · Nature Communications]

Reviewers' comments:

Reviewer #1 (Remarks to the Author):

The manuscript titled "Real-time, non-contact, cellular imaging and angiography of human cornea and limbus with common-path Full-field/SD OCT" by Viacheslav Mazlin, et.al, describes a combined Full-Field OCT (FF-OCT) and Spectral Domain OCT (SD-OCT) system for imaging the cellular structure of the human cornea and limbus in-vivo and without contact with the tissue surface.

The optical design of the FF-SD-OCT system is novel. It allows for tracking the focal plane and therefore for optimizing the OCT signal and improving the SNR of the OCT images. Although in-vivo imaging of the human cornea with the FF-OCT technology has already been published, to the best of my knowledge, this is the first time FF-OCT has been used to image in-vivo the human limbus.

Although the manuscript is written very well and provides great detail about both the FF-OCT technology and its application for in-vivo non-contact imaging of the human cornea and limbus, there are a few statements that are questionable and a few details that are missing. Below is a list of questions and suggested revisions.

1. Abstract, Line 3 from top: The authors claim that this is the first time that the cellular structure of the human cornea and limbus were imaged with OCT in-vivo and without contact with the tissue surface. This statement is incorrect, as recently there have been 2 publications that have showed such cellular resolution images:

B. Tan, et. al., "250 kHz, 1.5 μm resolution SD-OCT for in-vivo cellular imaging of the human cornea", Biomed. Optics Express Vol. 9, p. 6569-6583 (2018).

K. Bizheva, B. Tan, B. MacLellan, Z. Hosseinae, E. Mason, D. Hileeto and L. Sorbara, "In-vivo imaging of the palisades of Vogt and the limbal crypts with sub-micrometer axial resolution Optical Coherence Tomography", Biomed. Optics Express Vol. 8, No. 9, 4141-4151 (2017).

Suggest that the authors revise the abstract to reflect this fact.

2. Page 5, Paragraph 1, line 3 from top: The authors state that the spatial resolution of the FF-OCT system is $1.7\mu\text{m} \times 7.7\mu\text{m}$ (lateral x axial). Could the authors clarify if those values are measured in free space or calculated for corneal tissue. What was the average refractive index for corneal tissue used for the calculation?

3. Pages 6 and 7: The authors have addressed defocus due to axial eye motion. However, the human eye moves in XYZ direction and also twists along the XZ and XY planes. Would be helpful for the authors to comment how such motion affects the FF-OCT image quality.

4. Page 8, last line: Why is the optical mismatch so different for imaging the central cornea, peripheral cornea and the limbus? Does this have to do with the corneal and limbal curvature and the fact that the FF-OCT resolution is not isotropic? Would be helpful for the authors add a comment in this section.

5. Page 10, Figure legend and page 11, paragraph 2, line 4: The authors state that: "...Resolving individual keratocyte nuclei (yellow arrows) was increasingly more difficult, when imaging further from the center...". Can the authors explain why? Is that relevant to the curvature of the cornea and the non-isotropic FF-OCT resolution?

6. Page 10, Paragraph 1, line 4 from top: Please, provide references to histology / anatomy data (citation of publications) that correlat the size of the corneal epithelial cells and their nuclei

obtained from the FF-OCT with histology.

7. Page 10, Paragraph 1, line 17 from top: The authors state that: "...With increasing depth, nuclei become more elongated and their density decreases...". Please, provide references to histology / anatomy data (citation of publications) that supports this statement.

8. Page 11, Paragraph 1, lines 1 and 2 from top: Although FF-OCT is immune to spherical and astigmatic aberrations, both spectrally-dependent scattering and absorption in biological tissue, that is also spatially-dependent, will affect the OCT axial resolution. The authors may want to revise this paragraph accordingly.

9. Page 13, "Discussion" Paragraph 1, lines 2-3 from top: The authors state that: "...Conventional OCT provides high axial resolution of corneal layers but does not resolve cells...". This statement is incorrect. Evidence to that fact is provided in the 2 references cited in item #1 on this list. Suggest that the authors rephrase the statement accordingly.

10. Page 14, Paragraph 1, lines 17-19 from top: The authors state that: "...Furthermore, our instrument is not limited to diagnosis of "static" corneal disorders, but can potentially monitor inflammatory and scarring conditions affecting the dynamics of the blood flow in the eye...". The way this statement is phrased, it suggests that only FF-OCT can do that, which is incorrect. Both IVCM and OCT technology can provide "dynamic" examination of biological tissue and monitor diseases over time. Suggest the authors rephrase this statement.

11. Page 15, Paragraph 2, lines 3 from top: What was the entrance aperture of the Olympus objective and was that aperture filled 100% in order to achieve 1.7 μm lateral resolution?

12. Page 15, Paragraph 2, lines 16-17 from top: The authors state that: "...Reflectivity of the cornea, estimated from the Fresnel relations, is around 2%...". This is probably true for the air/tear film interface, however, the reflection at the endothelium / anterior segment chamber should be < 2%.

Reviewer #2 (Remarks to the Author):

This well-written paper is very important for future clinical application of novel FF-OCT system.

1. Figure 4

Evaluation of endothelial cells at the central part of the cornea is essential for clinical application of FF-OCT. But, this paper (and ref 10) did not show the image of endothelial cells at the central cornea. Less influence by corneal curvature is expected at the central cornea, hence, central part of the cornea is ideal location to evaluate corneal tissue by en face imaging. Are there any essential problems in FF-OCT to evaluate the endothelial cells at the central cornea? If there are essential problems, authors should discuss about it.

2. Page 10. "we were able to perform clinically significant cell counting and measured the normal endothelial cell density of 3096 cells/mm²"

Comparison with the data from commercially available specular microscope is important to confirm it.

3. Figure 4 and text in page 11.

Imaging of sclera is difficult by signal attenuation with the light source (930 nm) of this FF-OCT. Images of the sclera by FF-OCT in Fig 4 was not originated from sclera, but from either Tenon's capsule, conjunctiva or episclera.

4. Imaging of in vivo human tear film

Authors describe the motion compensation for heartbeat and breathing. In the clinical retinal OCT (especially OCTA), motion compensation for saccadic eye movement is important to get good images with long measurement time. How did authors process the saccadic eye movement?

5. Page 12. "Moreover, using rapid 275 frames/s acquisition, we could visualize and track the flow of these cells,"

These cells are RBC or WBC?

Reviewer #3 (Remarks to the Author):

The manuscript proposes using a novel full field OCT system, with an additional commercial spectral domain OCT system to provide axial tracking, to image corneal cellular structures in vivo without being in contact with the sample. Axial tracking was tested using ex vivo macaque cornea following a physiologically inspired periodic motion profile. In vivo imaging was performed near the corneal apex and at the corneoscleral limbus. Angiography was performed by calculating the location and velocity of individual erythrocytes within blood vessels which provides a substantial advantage over other variance and Doppler OCT based angiography techniques.

Comments

- Movie 1 (corresponding to Figure 1) is useful in visualizing the simultaneous motion within the system. However, may I suggest moving some of the external components to be closer to one another so that the viewer may more easily track what is happening instead of it taking place throughout the entire image?

- Some of these cellular structures are very thin axially. Increasing the FFOCT source bandwidth would allow for this. Beyond the financial cost of a new source, are there unique engineering challenges in FFOCT that would prevent this?

- For real time imaging, is the system limited to only displaying 10 frames per second or is the acquisition rate also limited to 10 frames per second?

- At one point in the manuscript the voice coil motor is stated to operate at 50 Hz however in the methods section, the stage encoder is listed as being limited to 20 new positions/sec. Which one is it?

- For visualization of corneal epithelial and endothelial cells, would there be an advantage to applying both averaging and Fourier domain filtering to the images instead of only one to each?

- For imaging blood cells (Fig. 6m), a differential between two images is taken. The authors note that half of the useful signal is lost by performing this method. How so? It would remove static areas of the image but highlight those areas in motion which seems to be the signal of interest. Can the authors clarify this?

- The authors state that the corneal irradiance was measured to be below the maximal permissible exposure with an exposure level at 2% on the retina. The ANSI Z80.36 standard now offers calculations specifically for the cornea. What was the exposure MPE percentage for the cornea? And what were the individual contributions for the SD and FFOCT sources?

Results

- In the ex vivo defocus test, there is phase washout during acquisition. Why does this occur at only some points if the motion is at constant velocity? What are the phase washout limitations of FF-OCT with comparison to SD and SS OCT? And what is the limitation of your system in particular?

o Hansford C. Hendargo, Ryan P. McNabb, Al-Hafeez Dhalla, Neal Shepherd, and Joseph A. Izatt, "Doppler velocity detection limitations in spectrometer-based versus swept-source optical coherence tomography," Biomed. Opt. Express 2, 2175-2188 (2011)

- There appears to be an error in Fig. 3 C with the absolute error not corresponding to the top values.

- The authors state that FFOCT is insensitive to optical aberrations. While the authors have previously published on this phenomena, it may be useful to the reader to expand on this concept and the value it provides in this context. Also, are the spherical and astigmatic aberrations referred to in the manuscript text located within the FFOCT system or naturally occurring aberrations inherent to the cornea?

- Conventional SD and SSOCT systems capture single depth profiles and scan a point to build up a volume. Given the axial tracking and defocus corrections available in the described FFOCT system, can the authors generate 3D volumes from their data?

- Only axial tracking is described and tested. What about lateral tracking? Is there any way to compensate for a saccade or drift between acquisition frames?

- It would be useful to label the figures or figure captions to denote which subject the data came from. Ideally there would at least one direct comparison for a target structure for all three imaged volunteers.

- Can the authors comment on the high specular reflectivity at the boundary between the endothelium and aqueous humor? While there is a Purkinje reflection from that surface, it usually is not at the same intensity as the anterior surface of the cornea.

- I would encourage the authors to use an editing service to help improve the readability of the manuscript and the layout of the figures and movies.

Dear Reviewers,

We truly appreciate the careful and deeply analyzed reviews and thank for pointing out the novelty of our work.

Please find our responses to every specific comments below. The manuscript was carefully modified based on all suggestions.

Sincerely,
A. Claude BOCCARA

Reviewer #1 (Remarks to the Author):

The manuscript titled "Real-time, non-contact, cellular imaging and angiography of human cornea and limbus with common-path Full-field/SD OCT" by Viacheslav Mazlin, et.al, describes a combined Full-Filed OCT (FF-OCT) and Spectral Domain OCT (SD-OCT) system for imaging the cellular structure of the human cornea and limbus in-vivo and without contact with the tissue surface.

The optical design of the FF-SD-OCT system is novel. It allows for tracking the focal plane and therefore for optimizing the OCT signal and improving the SNR of the OCT images. Although in-vivo imaging of the human cornea with the FF-OCT technology has already been published, to the best of my knowledge, this is the first time FF-OCT has been used to image in-vivo the human limbus.

Although the manuscript is written very well and provides great detail about both the FF-OCT technology and its application for in-vivo non-contact imaging of the human cornea and limbus, there are a few statements that are questionable and a few details that are missing. Below is a list of questions and suggested revisions.

1. Abstract, Line 3 from top: The authors claim that this is the first time that the cellular structure of the human cornea and limbus were imaged with OCT in-vivo and without contact with the tissue surface. This statement is incorrect, as recently there have been 2 publications that have showed such cellular resolution images:

B. Tan, et. al., "250 kHz, 1.5 μm resolution SD-OCT for in-vivo cellular imaging of the human cornea", Biomed. Optics Express Vol. 9, p. 6569-6583 (2018).

K. Bizheva, B. Tan, B. MacLellan, Z. Hosseinae, E. Mason, D. Hileeto and L. Sorbara, "In-vivo imaging of the palisades of Vogt and the limbal crypts with sub-micrometer axial resolution Optical Coherence Tomography", Biomed. Optics Express Vol. 8, No. 9, 4141-4151 (2017).

In the abstract, we highlight the central problem – "***In today's clinics***, a cell-resolution view of the cornea can be achieved only with a confocal microscope (IVCM) in contact with the eye.". By limiting the scope ***only to clinical devices*** we make the problem in the abstract clear and accessible to the large audience and avoid cumbersome discussion about advantages and disadvantages of each of the research devices (UHR-OCT and our previous FFOCT), which we cover and cite in the main text body. (And truly, in hospitals today confocal microscopes are the only devices that can provide cell detail images of the cornea).

Nevertheless, in order to emphasize the importance of research works in the field of cell-resolution corneal imaging, we expand the introduction with discussion on FFOCT and UHR-OCT devices:

“This technology, originating from the lower speed *ex vivo* FFOCT^{11,12,13,14}, uses a 2D camera to acquire high-resolution *en face* images i) directly without beam scanning artifacts, ii) rapidly (at a camera frame rate), iii) with flexible FOV. These qualities distinguish FFOCT from another high-resolution modality - UHR-OCT, which also achieved impressive cell-detail imaging in the cornea and limbus using a conventional scanning approach^{15,16}.”

11. Beaurepaire, E., Boccara, A. C., Lebec, M., Blanchot, L. & Saint-Jalmes, H. Full-field optical coherence microscopy. *Opt. Lett.* 23, 244 (1998).

12. Dubois, A. *Handbook of Full-Field Optical Coherence Microscopy: Technology and Applications.* (Pan Stanford publishing, 2016).

13. Grieve, K. et al. In vivo anterior segment imaging in the rat eye with high speed white light full-field optical coherence tomography. *Opt. Express* 13, 6286 (2005).

14. Grieve, K. et al. Ocular Tissue Imaging Using Ultrahigh-Resolution, Full-Field Optical Coherence Tomography. *Investig. Ophthalmology Vis. Sci.* 45, 4126 (2004).

15. Tan, B. et al. 250 kHz, 1,5 μm resolution SD-OCT for in-vivo cellular imaging of the human cornea. *Biomed. Opt. Express* 9, 6569 (2018).

16. Bizheva, K. et al. In-vivo imaging of the palisades of Vogt and the limbal crypts with sub-micrometer axial resolution optical coherence tomography. *Biomed. Opt. Express* 8, 4141 (2017).

2. Page 5, Paragraph 1, line 3 from top: The authors state that the spatial resolution of the FF-OCT system is $1.7\mu\text{m} \times 7.7\mu\text{m}$ (lateral x axial). Could the authors clarify if those values are measured in free space or calculated for corneal tissue. What was the average refractive index for corneal tissue used for the calculation?

Lateral resolution was estimated from the Rayleigh criterion:

$$\Delta x = \Delta y = \frac{0.61\lambda}{NA} = \frac{0.61 \cdot 850\text{nm}}{0.3} = 1.7\ \mu\text{m},$$

where λ is the central wavelength of the light source and NA is the numerical aperture of the detection. Due to the Snell's law, the numerical aperture (and, therefore, the lateral resolution) is the same in free space and in the cornea.:

$$NA_{air} = n_{air} \cdot \sin \theta_{air} = n_{cornea} \cdot \sin \theta_{cornea} = NA_{cornea}$$

Experimentally lateral resolution was first measured using the 1951 USAF target, however even the smallest lines were clearly resolved, confirming that the FFOCT resolution is better than $2.19\ \mu\text{m}$. In order to get the precise experimental confirmation of the fine $1.7\ \mu\text{m}$ resolution, we measured the visible diameters (FWHM) of 80 nm gold nanoparticles on a glass plate, located in the focus of FFOCT objective. The target and the measured plot of the amplitude profile across the particle are shown below in Additional Fig.1.

Additional Fig. 1 Experimental measurements of lateral resolution.

The axial resolution was estimated in the cornea from the experimentally measured FWHM 30 nm spectral bandwidth of the LED (using spectrometer (CCS175/M, Thorlabs, USA)), shown below in Additional Fig.2 and using the average refractive index of the cornea $n=1.376$, often quoted in the literature (Patel, S. and Tutchenko, L. (2019). The refractive index of the human cornea: A review. Contact Lens and Anterior Eye, 42(5), pp.575-580.).

$$\Delta z = \frac{2 \ln(2)}{\pi \cdot n} \frac{\lambda^2}{\Delta \lambda} = \frac{2 \ln(2)}{\pi \cdot 1.376} \frac{(850 \text{ nm})^2}{30 \text{ nm}} \approx 7.7 \mu\text{m}$$

Additional Fig. 2 Experimental measurement of light source (LED) bandwidth, affecting axial resolution.

The above information is added to the Main text and the Methods section:

Main text:

“The FFOCT interferometer, equipped with a near-infrared (NIR) 850 nm incoherent light-emitting diode (LED) source and moderate numerical aperture (0.3 NA) 10× air microscope objectives (MO), acquires 2D *en face* 1.25 mm × 1.25 mm images of XY corneal sections with 1.7 μm lateral and 7.7 μm axial resolutions (see Methods) by a time-domain two-phase shifting scheme¹² (**Fig. 1, Supplementary Videos 1,2**).”

Methods:

“The FFOCT device is based on an interference microscope in a Linnik configuration with identical microscope objectives in the two arms of the interferometer. Objectives (LMPLN10XIR, Olympus, Japan) have a numerical aperture (NA) of 0.3, 10× magnification and give high lateral resolution of 1.7 μm (with a filled 10.8 mm entrance aperture). The lateral resolution (identical in cornea and in free space) was estimated according to the Rayleigh criterion and experimentally confirmed with a resolution target and by measuring the diameters (FWHM) of 80 nm gold nanoparticles imaged on a glass plate, located at the focal plane of the FFOCT objective.”

“The axial resolution of 7.7 μm in the cornea was estimated from the experimentally measured spectral bandwidth of the LED (30 nm) with spectrometer (CCS175/M, Thorlabs, USA) and by using the average corneal refractive index of 1.376, often quoted in the literature⁴⁶.”

46. Patel, S. & Tutchenko, L. The refractive index of the human cornea: A review. *Contact Lens Anterior Eye* **42**, 575–580 (2019).

3. Pages 6 and 7: The authors have addressed defocus due to axial eye motion. However, the human eye moves in XYZ direction and also twists along the XZ and XY planes. Would be helpful for the authors to comment how such motion affects the FF-OCT image quality.

We thank reviewer for the relevant question. It is helpful to distinguish the two effects: i) shift of the eye position as a consequence of eye motion and ii) overall rapid eye motion itself.

The mentioned problem of defocus is caused only by the first effect – shifts of the eye position and corresponding shifts of the focus plane and coherence planes inside the eye (refractive index of the cornea and Snell’s law). As mentioned in the text of the article, the Z-shift of the focus plane is corrected automatically with defocusing correction procedure. The X- and Y-shifts are typically much slower than the Z-shifts (which have high frequencies due to heartbeat), therefore we correct them manually by moving the whole device with XY motorized stages using the joystick.

The second effect – the rapid eye motion – affects the FF-OCT as well. The rapid Z-motion of the eye introduces additional phase modulation, which, being added to the π piezo modulation, reduces the FF-OCT signal. The effect of rapid Z-motion is seen in FF-OCT videos by occasionally vanishing tomographic signal (see Additional Fig. 3 below).

Additional Fig. 3 Effect of rapid Z-motion of the eye. Both images are taken from the same corneal layer, but the right image has reduced FF-OCT signal due to the presence of the rapid axial motion.

The rapid saccadic XY motion of the eye as well as the rapid flow of tears after a blink, can also introduce artifacts to the FF-OCT images. More precisely, in the FF-OCT scheme camera captures the light from the entire thickness of the sample, while the optical interference fringes are located only in a single sample layer (which matches with reference mirror in terms of optical path length). In order to extract only the light from this single layer, the camera captures the two images with fringes shifted by π (opposite contrast) and images are subtracted. In case of static sample, all the light outside of the coherence fringe volume will be removed (because it is the same in both images and not affected by the π shift), while the light from the single sample layer with fringes will be doubled. In case of the laterally moving sample, the above subtraction will not completely remove the light outside of the coherence volume, because the scene is shifted. As the largest proportion of out-of-coherence-volume light is coming from the surface of the eye (due to large difference between the air-tear film refractive indexes), we will see the artifacts that manifest the defocused view of the ocular surface (Additional Fig. 4). Those artifacts are occasionally visible in the moments of saccadic motion or right after the blink of the eye, when the tear flow is fast. The artifacts are less present, when imaging the deeper corneal layers (deep stroma, endothelium), as the surface gets further out of focus of the optical system.

Additional Fig. 4 Effect of rapid XY-motion of the eye. Both images are taken from the same corneal layer, but right image contains the defocused artifacts from the corneal surface due to the presence of the saccadic motion.

We make the following additions to the Main text:

We add the Figure below to explain the process of defocus correction. This Figure clarifies that the problem of defocus is not connected with the rapid axial motion, but is connected only with the axial shift of the eye as a consequence of eye movements.

Fig. 3 Principle of defocus correction for matching the optical arms of FFOCT interferometer. a FFOCT interferometer is focused on the surface of the sample (blue square). The focus spot matches with the coherence plane, FFOCT image from the surface is captured and defocus

correction is not needed. **b** FFOCT interferometer is focused inside the sample. Due to Snell's law the focus is extended. At the same time the coherence plane is shifted in the opposite direction due to the higher refractive index inside the sample than in the air. The focus spot and the coherence plane are mismatched and FFOCT image cannot be captured without defocus correction. **c** FFOCT interferometer is focused inside the sample and the focus is extended. The Defocus correction procedure is performed: the reference arm is extended to put the coherence plane in the new location of the focus and the FFOCT image from inside of the sample is captured.

“It should be noted that ocular movements not only shift the position of the eye leading to the problem of defocus, but they also intervene in the FFOCT image retrieval scheme, affecting image quality. The rapid axial motion of the eye can introduce additional phase modulation, which, when added to the π piezo modulation, reduces the FFOCT signal. This effect is seen in FFOCT videos as occasional wash out of the tomographic signal (**Supplementary videos 6,7,8,9**). The rapid saccadic lateral motion of the eye as well as the rapid flow of tears after a blink can also introduce the artifacts to the FFOCT images. More precisely, the two-phase retrieval scheme is unable to completely remove the light originating from the outside of the coherence volume, because the scene in the two images is shifted. As the largest proportion of out-of-coherence-volume light comes from the air-tear film interface, the artifacts manifest the defocused view of the ocular surface. The artifacts occasionally appear in the FFOCT videos (**Supplementary videos 6,7,8,9**) and are more seldom present when imaging the deeper corneal layers (deep stroma, endothelium), as the surface gets further from the focus of the optical system.”

3 Page 8, last line: Why is the optical mismatch so different for imaging the central cornea, peripheral cornea and the limbus? Does this have to do with the corneal and limbal curvature and the fact that the FF-OCT resolution is not isotropic? Would be helpful for the authors add a comment in this section.

The reviewer refers to the last line on Page 8, which provides the numbers for average optical mismatch, when imaging the central cornea, peripheral cornea and sclera:

“optical mismatch with $9.4 \pm 6.2 \mu\text{m}$, $11.3 \pm 7.2 \mu\text{m}$ and $7.2 \pm 6.6 \mu\text{m}$ (mean \pm s.d.) errors, respectively”

The above numbers are close, as they are lying within the s.d. error margins of each other. Nevertheless, the small variability of the above numbers is attributed to the slightly different eye movement patterns of the subject during the exam (for example, longer experiment may lead to the increase of eye movements and slightly larger error in defocusing correction).

5. Page 10, Figure legend and page 11, paragraph 2, line 4: The authors state that: “...Resolving individual keratocyte nuclei (yellow arrows) was increasingly more difficult, when imaging further from the center...”. Can the authors explain why? Is that relevant to the curvature of the cornea and the non-isotropic FF-OCT resolution?

We provide the necessary clarification:

Caption of Figure 6:

“Resolving individual keratocyte nuclei (yellow arrows) was increasingly more difficult, when imaging further from the center, **due to stronger light scattering from the stromal fibrils and consequent glare over the entire image.**”

Main text:

“We also looked at the appearance of stroma in central and peripheral cornea, and sclera. The dark background of central corneal stroma (**Fig. 6d**), becomes bright at the periphery (**Fig. 6e,f**), which is explained by the increased light scattering from the stromal fibrils, irregular in diameter and arrangement, known from electron microscopy studies³⁴. **The above scattering bright glare of the background makes it more difficult to resolve keratocyte cell nuclei at the periphery, comparing to those in central cornea easily visible against the dark background, in agreement with previous confocal microscopy data³⁵.**”

35. Zheng, T., Le, Q., Hong, J. & Xu, J. Comparison of human corneal cell density by age and corneal location: an in vivo confocal microscopy study. *BMC Ophthalmol.* **16**, (2016).

6. Page 10, Paragraph 1, line 4 from top: Please, provide references to histology / anatomy data (citation of publications) that correlate the size of the corneal epithelial cells and their nuclei obtained from the FF-OCT with histology.

We provided the necessary reference:

“Just beneath the tear film, we could see superficial epithelial cells 40 - 50 μm in diameter with dark 8 - 13 μm nuclei, in agreement with literature²³ (**Fig. 6b,g**).”

23. Guthoff, R. F., Baudouin, C. & Stave, J. *Atlas of Confocal Laser Scanning In-vivo Microscopy in Ophthalmology*. (Springer Berlin Heidelberg, 2006). doi:10.1007/3-540-32707-X.

Furthermore, we obtained the superficial epithelial images from a healthy volunteer using the clinical confocal microscope (CM) in contact with the eye (HRT II with Rostock cornea module; Heidelberg Engineering, GmbH, Germany). The comparison between the entire CM image (field of view = 0.3 mm x 0.3 mm) and zoomed FFOCT image of the same size, confirms identical visible diameters of the cells and nuclei (Additional Fig. 5).

Additional Fig. 4 Comparison of superficial cell and nuclei sizes between the CM and zoomed FFOCT images. FOV = 0.3 mm x 0.3 mm.

7. Page 10, Paragraph 1, line 17 from top: The authors state that: "...With increasing depth, nuclei become more elongated and their density decreases...". Please, provide references to histology / anatomy data (citation of publications) that supports this statement.

Additional Fig. 5 FFOCT images from different stromal depths.

We precise our statement and provided the necessary references:

"From the anterior to posterior stroma the density of keratocytes gradually decreases down to the deep stromal layer, adjacent to Descemet's membrane, where the keratocyte density shows a slight increase, in agreement with literature²⁸. Moreover, with the increasing depth, the nuclei show a more elongated shape²⁹ (Fig. 5k,l)."

28. Berlau, J., Becker, H.-H., Stave, J., Oriwol, C. & Guthoff, R. F. Depth and age-dependent distribution of keratocytes in healthy human corneas: a study using scanning-slit confocal microscopy in vivo. *J. Cataract Refract. Surg.* **28**, 611–616 (2002).

29. Darlene A. Dartt *et al.* *Ocular Periphery and Disorders*. (Academic Press, 2011).

8. Page 11, Paragraph 1, lines 1 and 2 from top: Although FF-OCT is immune to spherical and astigmatic aberrations, both spectrally-dependent scattering and absorption in biological tissue, that is also spatially-dependent, will affect the OCT axial resolution. The authors may want to revise this paragraph accordingly.

We thank the reviewer for the comment and revise the above paragraph:

"Nonetheless, it should be noted that both spectrally-dependent scattering and absorption in biological tissue, that is also spatially-dependent, do affect the FFOCT axial resolution."

9. Page 13, "Discussion" Paragraph 1, lines 2-3 from top: The authors state that: "...Conventional OCT provides high axial resolution of corneal layers but does not resolve cells...". This statement is incorrect. Evidence to that

fact is provided in the 2 references cited in item #1 on this list. Suggest that the authors rephrase the statement accordingly.

We rephrase our statement to encompass the UHR-OCT, which can resolve the cells:

“Conventional OCT provides high axial resolution of corneal layers, but does not resolve cells, while UHR-OCT resolves cells, however the *en face* images suffer from eye movement artifacts during X-Y beam scanning.”

10. Page 14, Paragraph 1, lines 17-19 from top: The authors state that: “...Furthermore, our instrument is not limited to diagnosis of “static” corneal disorders, but can potentially monitor inflammatory and scarring conditions affecting the dynamics of the blood flow in the eye...”. The way this statement is phrased, it suggests that only FF-OCT can do that, which is incorrect. Both IVCN and OCT technology can provide “dynamic” examination of biological tissue and monitor diseases over time. Suggest the authors rephrase this statement.

We thank the reviewer for the comment, although we would like to clarify that, when we spoke about the *inflammatory and scarring conditions affecting the dynamics of the blood flow in the eye*, we meant very fast processes (on a time scale of milliseconds), for example propagation of inflammatory cells through the vessel and not just an hour-to-hour follow-up. Although, IVCN and OCT demonstrated static images of the blood flow, they have insufficient speed or resolution to follow the propagation of the individual cells through the vessel – while FFOCT, thanks to its fast 0.6 billion pixels/second (*en face*) acquisition speed can perform such monitoring.

We highlight in the text that FFOCT is unique only in monitoring the fast (milliseconds) dynamic conditions (propagation of cells in the blood vessels of anterior eye):

“Furthermore, our instrument is not limited to diagnosis of “static” corneal disorders, but can potentially monitor inflammatory and scar conditions affecting the **fast (millisecond)** dynamics of the blood flow in the eye.”

11. Page 15, Paragraph 2, lines 3 from top: What was the entrance aperture of the Olympus objective and was that aperture filled 100% in order to achieve 1.7 μm lateral resolution?

The entrance aperture of the Olympus objective measured 10.8 mm and was filled to achieve the 1.7 μm lateral resolution.

We add the answer to the Methods section:

“The FFOCT device is based on an interference microscope in a Linnik configuration with identical microscope objectives in the two arms of the interferometer. Objectives (LMPLN10XIR, Olympus, Japan) have a numerical aperture (NA) of 0.3, 10 \times magnification and give high lateral resolution of 1.7 μm (**with a filled 10.8 mm entrance aperture**).”

12. Page 15, Paragraph 2, lines 16-17 from top: The authors state that: "...Reflectivity of the cornea, estimated from the Fresnel relations, is around 2%...". This is probably true for the air/tear film interface, however, the reflection at the endothelium / anterior segment chamber should be $< 2\%$.

We agree with the reviewer that the reflection from endothelium/anterior chamber is well below 2%. However, according to the theory of FFOCT (Dubois, A. Handbook of Full-Field Optical Coherence Microscopy: Technology and Applications) signal-to-noise ratio is the largest, when the reflectivity from the reference mirror equals the **total** reflectivity of the sample (**from all the layers participating to the FFOCT signal or not**), and not only to the temporal coherent part originating from one slice. The total corneal reflectivity (air-tear film, tear film – superficial epithelium, endothelium – anterior chamber) is about 2%, therefore use of close - 4% reflectivity reference mirror gives us close to ideal performance in terms of SNR.

We clarify in the text that particularly the **total** sample reflectivity is important:

“A low reflectivity value is chosen to achieve high detection sensitivity, which is maximized when the total reflectivity (from all the layers participating or not to FFOCT signal) of the sample and reflectivity of the reference mirror match, as indicated by FFOCT signal to noise calculations¹². Total reflectivity from all corneal layers, estimated from the Fresnel relations, is around 2%. By using a reference mirror with a reflectivity of 4%, we can expect sensitivity close to the ideal condition¹².”

Reviewer #2 (Remarks to the Author):

This well-written paper is very important for future clinical application of novel FF-OCT system.

1. Figure 4

Evaluation of endothelial cells at the central part of the cornea is essential for clinical application of FF-OCT. But, this paper (and ref 10) did not show the image of endothelial cells at the central cornea. Less influence by corneal curvature is expected at the central cornea, hence, central part of the cornea is ideal location to evaluate corneal tissue by en face imaging. Are there any essential problems in FF-OCT to evaluate the endothelial cells at the central cornea? If there are essential problems, authors should discuss about it.

In fact, Figure 4 (Figure 5 in the revised manuscript) demonstrates endothelium in the central cornea – just 0.5 mm from the corneal apex (according to the accepted definition, central cornea is a circle of 2 mm radius around the corneal apex).

Nevertheless, below we provide the image that confirms that FFOCT can obtain endothelial images exactly in the corneal apex as well. Although, we could see the endothelial cell mosaic, we were unable to completely filter the interference fringes, because their spatial frequency was varying in a large range. The endothelium on the apex had the smallest fringe frequency, because its orientation was perpendicular to the optical axis – the same as the reference mirror. However, the endothelium located further from the corneal apex had a tilt relatively to the optical axis (and the reference mirror), which led to the visible increase in the interference fringe frequency.

Additional Fig. 6 FFOCT images from the central endothelium. The image on the right captures exactly the corneal endothelium apex.

In the text we highlight the central origin of the endothelial image. In the Methods we discuss the problem of varying interference fringe frequency, when imaging exactly on the corneal apex:

Main text:

“**Central** endothelium viewed in a single FFOCT image was hindered by a strong specular reflection and interference fringe artifacts; nevertheless, after lateral image registration and averaging (see Methods) we could resolve the hexagonal mosaic of 20 μm diameter cells and sometimes a 5 μm nucleus (**Fig. 5d,m**).”

Methods:

It is worth noting that filtering the interference fringes was more difficult in cases, when endothelium was captured exactly on the corneal apex, because the fringes had varying spatial frequency and spacing. The endothelium on the apex had the smallest fringe frequency, because its orientation was perpendicular to the optical axis, i.e. – the same orientation as the reference mirror. The endothelium elsewhere than at the apex was tilted due to the corneal curvature, which increased the interference fringe frequency.

2. Page 10. “we were able to perform clinically significant cell counting and measured the normal endothelial cell density of 3096 cells/mm²” Comparison with the data from commercially available specular microscope is important to confirm it.

We followed the advice of the reviewer and screened the same subject in the hospital using the commercial specular microscope (SP-3000P; Topcon, Japan). The endothelium image was obtained

from the central corneal region. Cells were counted in the area with the largest contrast and showed similar density (3100 cells/mm²) to FFOCT (3096 cells/mm²).

Additional Fig. 7 Clinical specular microscope image of endothelium

$$\text{Cell density} = \frac{310 \text{ cells}}{0.25 \text{ mm} \cdot 0.4 \text{ mm}} = 3100 \frac{\text{cells}}{\text{mm}^2}$$

We add the information to the text:

Main text:

We were able to perform clinically significant³⁰ cell counting and measured the normal endothelial cell density of 3096 cells/mm² (**Fig. 5d**), in agreement with the literature³¹, **and confirmed on the same subject using a clinical specular microscope, which counted 3100 cells/mm².**

Acknowledgements

We are grateful to Cristina Georgeon **and Marie Borderie** for assistance with acquiring the slit-lamp **and specular microscope** images.

Methods:

For comparison with FFOCT images endothelium was photographed using a clinical specular microscope (SP-3000P, Topcon, Japan) with 0.25 x 0.5 mm field of view.

3. Figure 4 and text in page 11. Imaging of sclera is difficult by signal attenuation with the light source (930 nm) of this FF-OCT. Images of the sclera by FF-OCT in Fig 4 was not originated from sclera, but from either Tenon's capsule, conjunctiva or episclera.

Indeed, the images of FFOCT were originating from episclera (**Fig. 6g**) and the upper portion of the sclera (**Fig. 6h**), where we saw only the shadows of overlaying conjunctival and episcleral vessels. Imaging the upper sclera is possible as penetration depth of FFOCT (according to the FFOCT theory) is comparable to that of other OCT techniques.

We correct the text accordingly:

Caption Figure 6:

Fig6 Common-path FF/SD OCT imaging of peripheral human cornea, episclera and upper sclera *in vivo*

Main text:

Blood vessels (**Fig. 6g**) were perforating the conjunctiva and episclera and produced shadows (**Fig. 6h**) in the upper sections of the sclera.

4. Imaging of *in vivo* human tear film. Authors describe the motion compensation for heartbeat and breathing. In the clinical retinal OCT (especially OCTA), motion compensation for saccadic eye movement is important to get good images with long measurement time. How did authors process the saccadic eye movement?

Due to the very fast (*en face*) acquisition speed of 0.6 billion pixels/second (that is achieved without beam scanning using a 2D camera), there are no beam scanning artifacts typical for classical OCT and OCTA, therefore we did not perform any lateral motion compensation.

However, it should be mentioned that FF-OCT has other types of artifacts, caused by the rapid saccadic XY motion of the eye as well as the rapid flow of the tears after the blink. More precisely, in the FF-OCT scheme the camera captures the light from the entire thickness of the sample, while the optical interference fringes are located only in a single sample layer. In order to extract only the light from this single layer, the camera captures the two images with fringes shifted by π (opposite contrast) and images are subtracted. In case of the static sample, all the light outside of the coherence fringe volume will be removed (because it is the same in both images and not affected by the π shift), while the light from the single sample layer with fringes will be doubled. In case of the laterally moving sample, the above subtraction will not completely remove the light outside of the coherence volume, because the scene is shifted. As the largest proportion of out-of-coherence-volume light is coming from the surface of the eye (due to big difference between the air-tear film refractive indexes), we will see the artifacts that manifest the defocused view of the ocular surface (Additional Fig. 8). Those artifacts are occasionally visible in the moments of saccadic motion or right after the blink of the eye, when the tear flow is fast. The artifacts are less present, when imaging the deeper corneal layers (deep stroma, endothelium), as the surface gets further out of focus of the optical system.

Additional Fig. 8 Effect of rapid XY-motion of the eye. Both images are taken from the same corneal layer, but right image contains the defocused artifacts from the corneal surface due to the presence of the saccadic motion.

For tear film imaging, the above defocused surface artifacts are irrelevant, because imaging was performed in the usual microscope configuration (FF-OCT with blocked reference arm). Moreover, in a usual microscope configuration images can be captured twice faster (1.75 ms per frame at 550 frames/second) than in FF-OCT, as there is no need for 2-phase retrieval scheme, therefore the lateral motion of the sample is completely frozen during the acquisition of one frame.

We make the following additions to the Main text:

“The rapid saccadic lateral motion of the eye as well as the rapid flow of tears after a blink can also introduce the artifacts to the FFOCT images. More precisely, the two-phase retrieval scheme is unable to completely remove the light originating from the outside of the coherence volume, because the scene in the two images is shifted. As the largest proportion of out-of-coherence-volume light comes from the air-tear film interface, the artifacts manifest the defocused view of the ocular surface. The artifacts occasionally appear in the FFOCT videos (**Supplementary videos 6,7,8,9**) and are more seldom present when imaging the deeper corneal layers (deep stroma, endothelium), as the surface gets further from the focus of the optical system.”

5. Page 12. “Moreover, using rapid 275 frames/s acquisition, we could visualize and track the flow of these cells,”
These cells are RBC or WBC?

Although, in the thin vessel we saw the particles propagating one after another and measured the 7 μm particle diameter, which corresponds to RBC, we were yet unable to distinguish between RBC and WBC in a general case of thick vessels, due to low contrast between the neighboring cells. Therefore, in the text we keep our statement restricted to “blood cells”.

Reviewer #3 (Remarks to the Author):

The manuscript proposes using a novel full field OCT system, with an additional commercial spectral domain OCT system to provide axial tracking, to image corneal cellular structures in vivo without being in contact with the sample. Axial tracking was tested using ex vivo macaque cornea following a physiologically inspired periodic motion profile. In vivo imaging was performed near the corneal apex and at the corneoscleral limbus. Angiography was performed by calculating the location and velocity of individual erythrocytes within blood vessels which provides a substantial advantage over other variance and Doppler OCT based angiography techniques.

Comments

- Movie 1 (corresponding to Figure 1) is useful in visualizing the simultaneous motion within the system. However, may I suggest moving some of the external components to be closer to one another so that the viewer may more easily track what is happening instead of it taking place throughout the entire image?

We followed the recommendation of reviewer and made one more additional Movie (Supplementary video - 3D scheme of setup for eye tracking and defocusing correction (close view, optics only)).

This movie demonstrates the optics of the FFOCT / SDOCT device in a close view:

Additional Fig. 9 One frame from a new Supplementary video - 3D scheme of setup for eye tracking and defocusing correction (close view, optics only)

- Some of these cellular structures are very thin axially. Increasing the FFOCT source bandwidth would allow for this. Beyond the financial cost of a new source, are there unique engineering challenges in FFOCT that would prevent this?

We agree with the reviewer that increasing the spectral bandwidth of the FFOCT source would increase the axial resolution of FFOCT. Unfortunately, there are no Broadband NIR LED sources of sufficient power available up to this moment (according to our knowledge). We are particularly interested in LEDs among all spatially and temporary incoherent sources, because they can be easily driven in pulsation mode (which is important to be compliant with ISO 15004-2 ocular safety). High power is needed to saturate the high-full well capacity camera (2 Mē), while having only 2% reflection from the eye and 4% reflection from the reference mirror.

Nevertheless, even with the current technology it is possible to combine several NIR LEDs with close spectral characteristics using a dichroic mirror and therefore increase the axial resolution.

We add this explanation to the main text:

“Axial resolution could be improved (for resolving finer corneal layers) in the future iterations of the device through increasing the spectral bandwidth of the source by implementing emerging broadband NIR LEDs of sufficient power or by combining several existing NIR LEDs with close spectral characteristics.”

- For real time imaging, is the system limited to only displaying 10 frames per second or is the acquisition rate also limited to 10 frames per second?

While our system is strongly optimized for both fast acquisition rate and simultaneous display at 275 FFOCT images/s (without skipping any frames), we artificially lowered both acquisition and display rate to 10 frames/second in order to be compliant with European ISO 15004-2 (2007) ocular safety standard. 10 fast pulsed exposures/s (3.5 ms each exposure) with pauses (no light) in between each exposure allow us to increase the output of the LED to saturate the camera, while still being below the maximum permissible light levels.

It should be noted that both the most recent US ocular safety standard ANSI Z80.36-2016 and the new European ISO standard (currently in draft stage and will be coming in this year) impose much less strict limitation for exposure of the anterior eye, which enables the use of our device at full speed frame rate at 275 frames per second without LED pulsation.

We add this explanation to the main text and methods:

Main text:

“While the system was heavily optimized for both fast acquisition rate and simultaneous display at 275 FFOCT images/s, we artificially lowered both acquisition and display rates of real-time imaging to 10 frames/s (with each frame captured in 3.5 ms) in order to be compliant with ocular safety standards.”

Methods:

“The real-time mode has artificially lowered acquisition and display rate to 10 frames/s in order to be compliant with European ISO 15004-2:2007 ocular safety standard (see below). 10 fast pulsed LED exposures/s (3.5 ms each exposure) with pauses in between each exposure allow us to increase the output of the LED to saturate the high full-well capacity camera, while still being below the maximum permissible light levels. It should be noted that the most recent US ocular safety standard ANSI Z80.36-2016 imposes a much less strict limitation for exposure of the anterior eye, which enables the use of our device at full speed frame rate at 275 frames per second without LED pulsation.”

- At one point in the manuscript the voice coil motor is stated to operate at 50 Hz however in the methods section, the stage encoder is listed as being limited to 20 new positions/sec. Which one is it?

We thank the reviewer for pointing on a typo and we correct it in the methods section:

Methods:

“The weak link in the communication is the limited read frequency of the stage encoder, which can accept **50 new positions/s** – about 2 times slower compared to the rate of provided positions by SDOCT.”

- For visualization of corneal epithelial and endothelial cells, would there be an advantage to applying both averaging and Fourier domain filtering to the images instead of only one to each?

We answer in the methods section:

“We were able to suppress them by averaging multiple (23) tomographic FFOCT images without Fourier domain conversion (**although averaging and Fourier domain processing can be used together to improve the final image**).”

- For imaging blood cells (Fig. 6m), a differential between two images is taken. The authors note that half of the useful signal is lost by performing this method. How so? It would remove static areas of the image but highlight those areas in motion which seems to be the signal of interest. Can the authors clarify this?

The image below is useful for understanding. When we retrieve the FFOCT image (2-phase method), we subtract two consecutive images on the camera and then take the absolute of the image to reveal the pixels that were hidden, due to their negative value (after subtraction). Unfortunately, in case of FFOCT imaging of the moving particle, we get the “doubling artifact” (we see the moving particle as the two objects separated by the small gap). This complicates particle tracking. To get a better view on the moving particles, we performed image subtraction without taking the absolute of the image. In this case each particle is seen as a single object, which simplifies its tracking, however half of the signal (negative pixels) is lost.

Additional Fig.10 Explanation of moving particle “doubling” in FFOCT and a proposed solution.

We add the explanation to the methods section:

“The image and video (**Fig. 8j, Supplementary video 12**) of individual blood cells was obtained not with a FFOCT image retrieval scheme, but by subtracting two consecutive images from the camera without taking the absolute value. The reason for this is the following: when we retrieve the FFOCT image via the two-phase method, the two consecutive images from the camera are subtracted and then the absolute of the image is taken to reveal the pixels that were hidden due to their negative value after subtraction. Unfortunately, in the case of FFOCT imaging of a moving particle, we get a “doubling artifact”, i.e. we see the moving particle as the two objects separated by a small gap. This complicates particle tracking. To get a better view of moving particles, we performed image subtraction without taking the absolute value. In this case each particle is seen as a single object, which simplifies its tracking. However, half of the signal (i.e. corresponding to the negative pixels) is lost.”

- The authors state that the corneal irradiance was measured to be below the maximal permissible exposure with an exposure level at 2% on the retina. The ANSI Z80.36 standard now offers calculations specifically for the cornea. What was the exposure MPE percentage for the cornea? And what were the individual contributions for the SD and FFOCT sources?

Initially, for the sake of simplicity in the article we mentioned only the average corneal exposure from LED and SLD together – 86 mW/cm², which is 86% of the ISO limit (100 mW/cm²) and 0.4% of the ANSI limit (20 W/cm²). (Such large difference between the two standards is due to the different limits that standards use for corneal exposure at 850 nm wavelength.)

Now we provide precise calculations that take into consideration that the exposures were pulsed and provide the exposure MPE percentages for cornea and retina in ISO and ANSI standards. The durations of pulsed light exposures were 3.5 ms for LED and 1.37 ms for SLD (for scanning beam the pulse duration is the time, during which the scanning beam is located within the diameter of the measurement aperture, specified by ISO and ANSI – 1 mm for anterior eye), durations of pauses without the light were 100 ms for LED and 0.41 ms for SLD and irradiances per pulse were 2 W/cm² for LED and 35 mW/cm² for SLD. These values were used to calculate the corneal exposure (example). The calculated time-dependent graphs (to be evaluated within a maximum time of 20 seconds, according to ISO and ANSI) show that during real-time acquisition we reach up to 52% of MPE for cornea and only 0.5% of MPE for retina (according to ISO 15004-2:2007) and 3.7% of MPE for cornea and 0.5% of MPE for retina (ANSI Z80.36-2016). The major contributor to the exposure is the LED.

Safety evaluation

Additional Fig. 11 Exposures of cornea (blue) and retina (green) through time (ISO standard). Safety limit is shown in red. Exposure level is normalized (see table).

Table. Pulsed safety evaluation at the highest exposure moments.	
FFOCT exposure (arbitrary units)	
$\left\{ \begin{array}{l} \frac{H_{IR-CL}^{850nm}}{1.8 \cdot t^{1/4}} \frac{J}{cm^2} + \frac{H_{IR-CL}^{930nm}}{1.8 \cdot t^{1/4}} \frac{J}{cm^2} < 1 \\ \frac{H_{VIR-R}^{850nm}}{\left(\frac{10}{d_r^{850nm}} \cdot t^{3/4}\right) \frac{J}{cm^2}} + \frac{H_{VIR-R}^{930nm}}{N^{-1/4} \cdot \left(\frac{10}{d_r^{930nm}} \cdot t^{3/4}\right) \frac{J}{cm^2}} < 1 \end{array} \right.$	Limit
$\left\{ \begin{array}{l} 0.52 \\ 0.005 \end{array} \right.$	$\left\{ \begin{array}{l} 1 \\ 1 \end{array} \right.$

We change the text accordingly:

Main text:

“The pulsed light irradiance was below the maximum permissible exposure (MPE) levels of up-to-date ISO 15004-2:2007 (52% of MPE for cornea and 0.5% of MPE for retina) and ANSI Z80.36-2016 (3.7% of MPE for cornea and 0.5% of MPE for retina) (for more details see Methods).”

Methods:

“A complete evaluation of corneal and retinal safety, was performed. In summary, we took into consideration the durations of pulsed light exposures (3.5 ms for LED and 1.37 ms for SLD (beam scanning)), durations of pauses without the light (100 ms for LED and 0.41 ms for SLD) and irradiances per pulse (2 W/cm² for LED and 35 mW/cm² for SLD). These values were used to calculate the corneal exposure (example). The calculated time-dependent graphs show that during real-time acquisition we reach up to 52% of MPE for cornea and only 0.5% of MPE for retina (according to ISO 15004-2:2007) and 3.7% of MPE for cornea and 0.5% of MPE for retina (ANSI Z80.36-2016). The major contributor to the exposure is the LED.”

- In the ex vivo defocus test, there is phase washout during acquisition. Why does this occur at only some points if the motion is at constant velocity? What are the phase washout limitations of FF-OCT with comparison to SD and SS OCT? And what is the limitation of your system in particular? o Hansford C. Hendargo, Ryan P. McNabb, Al-Hafeez Dhalla, Neal Shepherd, and Joseph A. Izatt, "Doppler velocity detection limitations in spectrometer-based versus swept-source optical coherence tomography," Biomed. Opt. Express 2, 2175-2188 (2011)

We thank reviewer for a relevant question. In fact, we found that the phase washout occasionally happened not only during the defocus test, when the sample was moving on a motorized stage at a constant velocity, but also, when the sample was completely static. This surprising effect was due to the unwanted small mechanical vibrations, caused by the external factors (e.g. underground metro passing close).

Nevertheless, to answer about the washout limitations of FFOCT: an unwanted additional π phase shift during 2 phase acquisition is sufficient to completely suppress the FFOCT signal. π phase shift for 850 nm equals $d = \frac{\phi\lambda}{2\pi} = 0.4 \mu m$. As the 2 images are acquired during the 3.5 ms, we come to the constant velocity (additional to the intentional π phase modulation) of about $0.4 \mu m / 3.5 ms = 0.1 mm/s$ between the arms of interferometer that is sufficient to suppress the FFOCT signal.

We add the necessary information to the main text:

Main text:

Only occasionally the signal vanished due to phase changes induced by unwanted mechanical vibrations, caused by external factors (e.g. underground metro passing close).

Figure caption:

FFOCT images are consistently acquired from various depths, while only occasionally the signal vanishes due to additional phase introduced to the tomographic signal by unwanted mechanical vibrations, caused by external factors (e.g. underground metro passing close) (Supplementary video 4)

- There appears to be an error in Fig. 3 C with the absolute error not corresponding to the top values.

We thank reviewer and substitute this figure with a correct one. We also correct the similar error in the next figure. Moreover, we adapt the figure formatting as recommended later by the same reviewer.

- The authors state that FFOCT is insensitive to optical aberrations. While the authors have previously published on this phenomena, it may be useful to the reader to expand on this concept and the value it provides in this context. Also, are the spherical and astigmatic aberrations referred to in the manuscript text located within the FFOCT system or naturally occurring aberrations inherent to the cornea?

We thank reviewer for the interesting question. We referred to the aberrations connected with the corneal curvature. In order to estimate the value that insensitivity of FFOCT resolution to optical aberrations brings to corneal imaging, we made a simulation in ZEMAX (see Additional Figure 12 below).

Simulation took into account the numerical aperture of 0.3, the field size of 1.25 mm and the anterior corneal curvature radius of 7.8 mm (The shape of the anterior and posterior surface of the aging human cornea, Vision Res. 2006 Mar; 46(6-7):993-1001). Effects of aberrations were measured at 550 μm depth, close to the back corneal side.

Cornea introduced mostly the spherical aberration and the field curvature. Spherical aberrations degrade the lateral resolution by 3 times to 5.1 μm (at the paraxial focus 550 μm deep inside the cornea). However, thanks to the immunity of FFOCT resolution to aberrations, we keep the diffraction-limited 1.7 μm resolution through the entire corneal thickness. This advantage should be even more apparent, when imaging pathological deformed corneas (e.g. keratoconus), which have more aberrations.

Aberrations in the image plane at the paraxial focus

Additional Fig. 12 Aberrations of the cornea, simulated with ZEMAX

We make the following addition:

Main text:

“We also benefited from the insensitivity of FFOCT to aberrations³². More precisely, the spatially incoherent light source in full-field illumination ensures that the non-aberrated light from the reference arm can effectively interfere only with a non-aberrated portion of light from the sample, while interference with the aberrated part is heavily suppressed³³. As a result, FFOCT keeps the diffraction-limited 1.7 μm resolution through the entire cornea, despite the presence of spherical aberrations, which are expected to reduce the resolution of conventional OCT and IVCM systems by three times to 5.1 μm (at the paraxial focus 550 μm deep inside the cornea).”

33. Xiao, P., Fink, M. & Boccara, A. C. Full-field spatially incoherent illumination interferometry: a spatial resolution almost insensitive to aberrations. *Opt. Lett.* 41, 3920 (2016).

Methods:

As FFOCT lateral resolution is immune to aberrations³³, the device is expected to resolve nerves and cells in pathological distorted corneas as well (e.g. case of keratoconus).

- Conventional SD and SSOCT systems capture single depth profiles and scan a point to build up a volume. Given the axial tracking and defocus corrections available in the described FFOCT system, can the authors generate 3D volumes from their data?

Indeed, we attempted to generate the 3D volumes from FFOCT images and SDOCT axial tracking data (see Figures of the whole cornea and of endothelium below). Yet, the quality of the 3D images was not sufficient for us to include them in the study. The reason is that FFOCT and SDOCT are driven by the 2 separate computers, which are synchronized with a NI-PSP protocol via the local network of the Langevin Institute. Although on average the delay between the computers in the network was measured to be 11 ms, this delay could vary in time, leading to the axial positions assigned to the wrong (earlier or later acquired) FFOCT images. Generation of 3D volume should be possible in the future iteration of the device, where both FFOCT and SDOCT are driven on the same PC.

Note: The defocusing correction did not suffer from the above network delays, because the same PC was driving the SDOCT axial tracking and the voice-coil motor of the FFOCT reference arm.

In vivo corneal volume

In vivo corneal endothelium
(composed of several depth slices)

We make the following additions to the Methods:

“SDOCT image acquisition is controlled by the separate PC, which is synchronized with FFOCT PC using the 11 ± 3 ms (mean \pm s.d.) precision NI-PSP protocol via the local network of the Langevin Institute. **Generation of 3D images should be possible in the future generations of the device with FFOCT and SDOCT being driven by a single PC.**”

- Only axial tracking is described and tested. What about lateral tracking? Is there any way to compensate for a saccade or drift between acquisition frames?

Yes, prior to averaging images of endothelium and revealing the hexagonal mosaic, we registered images with ImageJ plugin (Q. Tseng, E. Duchemin-Pelletier, A. Deshiere, M. Balland, H. Guillou, O. Filhol, and M. Théry, “Spatial organization of the extracellular matrix regulates cell-cell junction positioning,” Proc. Natl. Acad. Sci. U.S.A. 109(5), 1506–1511 (2012)) to compensate for the lateral movements.

We make the following additions to the main text:

“Central endothelium viewed in a single FFOCT image was hindered by a strong specular reflection and interference fringe artifacts; nevertheless, after **lateral image registration** and averaging (see Methods) we could resolve the hexagonal mosaic of 20 μ m diameter cells and sometimes a 5 μ m nucleus (**Fig. 5d,m**).”

We make the following additions to the methods:

“The same artifacts were also visible at the outer endothelial layer. We were able to suppress them by **registering and averaging multiple (23) tomographic FFOCT images with ImageJ**⁴⁷ without Fourier domain conversion (although averaging and Fourier domain processing can be used together to improve the final image).”

47. Tseng, Q. et al. Spatial organization of the extracellular matrix regulates cell-cell junction positioning. Proc. Natl. Acad. Sci. 109, 1506–1511 (2012).

- It would be useful to label the figures or figure captions to denote which subject the data came from. Ideally there would at least one direct comparison for a target structure for all three imaged volunteers.

The brief FFOCT image comparison between the three subjects can be found in our conference proceeding (Viacheslav Mazlin, Peng Xiao, Eugénie Dalimier, Kate Grieve, Kristina Irsch, José Sahel, Mathias Fink, Claude Boccara, "In vivo imaging through the entire thickness of human cornea by full-field optical coherence tomography," Proc. SPIE 10474, Ophthalmic Technologies XXVIII, 104740S.)

The next natural step is to collect images from a large cohort of normal subjects for comparison, however this was out of scope of the current study, which focused on demonstrating the technical feasibility of real-time FFOCT imaging.

- Can the authors comment on the high specular reflectivity at the boundary between the endothelium and aqueous humor? While there is a Purkinje reflection from that surface, it usually is not at the same intensity as the anterior surface of the cornea.

Indeed, theoretically the reflectivity from both boundaries is different by a factor of 100:

$$R_{air-tear\ film} = \frac{(n_{tear\ film} - n_{air})^2}{(n_{tear\ film} + n_{air})^2} = \frac{(1.336 - 1)^2}{(1.336 + 1)^2} \approx 0.02$$

$$R_{air-tear\ film} = \frac{(n_{endo} - n_{aqueous})^2}{(n_{endo} + n_{aqueous})^2} = \frac{(1.376 - 1.336)^2}{(1.376 + 1.336)^2} \approx 0.0002$$

From the images we see that the pixel values on the corneal surface and endothelium reflections are different by a factor of 10 (see the Figure below). However, contrary to the conventional OCT images of Intensity, the FFOCT acquires the images of Amplitude (due to 2-phase amplitude retrieval process), therefore the reflectivity estimated from images shows the same factor of 100, as theoretically expected:

$$\frac{R_{air-tear\ film}}{R_{endo-aqueous}} = \left(\frac{6000}{600} \right)^2 = 100$$

Surface reflection

Endothelial reflection

We make the following additions to the main text:

“While the reflectivity of endothelium-aqueous interface is expected to be 100 times less than air-tear film interface (based on the refractive index difference), FFOCT images show only 10 times difference in brightness, because FFOCT acquires the images of amplitude (due to 2-phase amplitude retrieval process) and not intensity like the conventional OCT.”

- I would encourage the authors to use an editing service to help improve the readability of the manuscript and the layout of the figures and movies.

In response to the reviewer’s comment concerning general readability, we have revised the whole manuscript text in detail with assistance from author and native English speaker KG. We have revisited all the figure layouts to improve clarity by re-organizing the figures, removing non-essential elements, increasing the sizes of the images within the figures and increasing the font size of the text to meet the publication format requirements of the journal. The updated figures are provided below:

REVIEWERS' COMMENTS:

Reviewer #1 (Remarks to the Author):

The authors have addressed adequately all of my concerns.

Reviewer #2 (Remarks to the Author):

Authors properly answer my questions.

Reviewer #3 (Remarks to the Author):

The thorough response to all initial comments were welcomed and the manuscript is much improved. I do have a few more comments:

- Some of the data in the manuscript (e.g. tear film visualization) was acquired without FFOCT enabled and instead the system performed optically as a conventional microscope. Was the SDOCT system and axial tracking enabled? If so this would be useful to the reader and would be a valuable differentiation from a conventional microscope.
- Regarding imaging speed and safety:
 - o It should be noted that the most recent US ocular safety standard ANSI Z80.36-2016 imposes a much less strict limitation for exposure of the anterior eye, which enables the use of our device at full speed frame rate at 275 frames per second without LED pulsation. – It should be made clear to the reader that this is just the potential of the system and that no imaging was performed at this rate (if my reading of the manuscript is correct).
 - o For the fast acquisition mode for blood flow visualization, are the LEDs pulsed at all or at constant illumination? How does that affect the power calculations? From the description in the methods it's not clear if this mode was outside the ISO safety limit.
- While the reflectivity of endothelium-aqueous interface is expected to be 100 times less than air-tear film interface... Because the figures in the manuscript do not show the raw amplitude values, I would make the additional comment that any appearance of similar brightness was due to adjustment to contrast or normalization.

Dear Reviewers,

Please find our responses to every specific comments below. The manuscript was carefully modified based on all suggestions from the last review stage.

Sincerely,
A. Claude BOCCARA

Reviewer #1 (Remarks to the Author):

The authors have addressed adequately all of my concerns.

Reviewer #2 (Remarks to the Author):

Authors properly answer my questions.

Reviewer #3 (Remarks to the Author):

The thorough response to all initial comments were welcomed and the manuscript is much improved. I do have a few more comments:

- Some of the data in the manuscript (e.g. tear film visualization) was acquired without FFOCT enabled and instead the system performed optically as a conventional microscope. Was the SDOCT system and axial tracking enabled? If so this would be useful to the reader and would be a valuable differentiation from a conventional microscope.

Although the axial tracking provides limited benefits, when imaging the tear film (because no defocus correction is required and the axial resolution of SDOCT is insufficient to resolve different layers of tear film), it can be still enabled for easier aligning to the eye.

We followed the recommendation of the reviewer and specified benefit of axial tracking of tear film imaging system:

“We were able to demonstrate tear film evolution by blocking the reference arm, thus converting our FF/SD OCT setup into a conventional microscope **with axial tracking**”

- Regarding imaging speed and safety: It should be noted that the most recent US ocular safety standard ANSI Z80.36-2016 imposes a much less strict limitation for exposure of the anterior eye, which enables the use of our device at full speed frame rate at 275 frames per second without LED pulsation. – It should be made clear to the reader that this is just the potential of the system and that no imaging was performed at this rate (if my reading of the manuscript is correct).

We highlighted that constant exposure in real-time imaging is possible only potentially in the future:

“It should be noted that the most recent US ocular safety standard ANSI Z80.36-2016 imposes a much less strict limitation for exposure of the anterior eye, which **in the future potentially** enables **prolonged** use of our device at full speed frame rate at 275 frames per second without LED pulsation.”

- For the fast acquisition mode for blood flow visualization, are the LEDs pulsed at all or at constant illumination? How does that affect the power calculations? From the description in the methods it's not clear if this mode was outside the ISO safety limit.

The blood flow imaging required constant illumination, but only during a short period of time: 40 camera frames (or 20 FFOCT images) equivalent to 70 ms. Following the advice of the reviewer we specify the safety calculations for this configuration as well:

In single pulse exposure scenario the durations of pulsed light exposures were 70 ms for LED and 1.37 ms for SLD (for scanning beam the pulse duration is the time, during which the scanning beam is located within the diameter of the measurement aperture, specified by ISO and ANSI – 1 mm for anterior eye), durations of pauses without the light were 30 s for LED and 0.41 ms for SLD and irradiances per pulse were 2 W/cm² for LED and 35 mW/cm² for SLD. These values were used to calculate the corneal exposure (example). The calculated time-dependent graphs (to be evaluated within a maximum time of 20 seconds, according to ISO and ANSI) show that during real-time acquisition we reach up to 18% of MPE for cornea and only 1% of MPE for retina (according to ISO 15004-2:2007) and 1.3% of MPE for cornea and 0.07% of MPE for retina (ANSI Z80.36-2016). The major contributor to the exposure is the LED.

Figure. Exposures of cornea (blue) and retina (green) through time. Safety limit is shown in red.

Table. Pulsed safety evaluation at the highest exposure moments (ISO 15004-2:2007)		
Scenario	FFOCT exposure (arbitrary units)	Limit

	$\left\{ \begin{array}{l} \frac{H_{IR-CL}^{850nm}}{1.8 \cdot t^{1/4}} \frac{J}{cm^2} + \frac{H_{IR-CL}^{930nm}}{1.8 \cdot t^{1/4}} \frac{J}{cm^2} < 1 \\ \frac{H_{VIR-R}^{850nm}}{\left(\frac{10}{d_r^{850nm}} \cdot t^{3/4}\right) \frac{J}{cm^2}} + \frac{H_{VIR-R}^{930nm}}{N^{-1/4} \cdot \left(\frac{10}{d_r^{930nm}} \cdot t^{3/4}\right) \frac{J}{cm^2}} < 1 \end{array} \right.$	
1	$\left\{ \begin{array}{l} 0.52 \\ 0.005 \end{array} \right.$	$\left\{ \begin{array}{l} 1 \\ 1 \end{array} \right.$
2	$\left\{ \begin{array}{l} 0.18 \\ 0.01 \end{array} \right.$	$\left\{ \begin{array}{l} 1 \\ 1 \end{array} \right.$

We change the text accordingly:

Methods:

“A complete evaluation of corneal and retinal safety, involving scenarios of real-time imaging (used to visualize static corneal structures) and single prolonged LED exposure (used to visualize blood flow dynamics), was performed. The results are summarized below.

For real-time imaging we took into consideration the durations of pulsed light exposures (3.5 ms for LED and 1.37 ms for SLD (beam scanning)), durations of pauses without the light (100 ms for LED and 0.41 ms for SLD) and irradiances per pulse (2 W/cm² for LED and 35 mW/cm² for SLD). These values were used to calculate the corneal and retinal exposures. The calculated time-dependent graphs show that during real-time acquisition we reach up to 52% of MPE for cornea and only 0.5% of MPE for retina (according to ISO 15004-2:2007) and 3.7% of MPE for cornea and 0.5% of MPE for retina (ANSI Z80.36-2016).

For single prolonged LED exposure we took into consideration the durations of pulsed light exposures (70 ms for LED and 1.37 ms for SLD (beam scanning)), durations of pauses without the light (30 s for LED and 0.41 ms for SLD) and irradiances per pulse (2 W/cm² for LED and 35 mW/cm² for SLD). The calculated time-dependent graphs show that during real-time acquisition we reach up to 18% of MPE for cornea and only 1% of MPE for retina (according to ISO 15004-2:2007) and 1.3% of MPE for cornea and 1% of MPE for retina (ANSI Z80.36-2016). The major contributor to the exposure is the LED.”

- While the reflectivity of endothelium-aqueous interface is expected to be 100 times less than air-tear film interface... Because the figures in the manuscript do not show the raw amplitude values, I would make the additional comment that any appearance of similar brightness was due to adjustment to contrast or normalization.

We follow the recommendation of the reviewer and add the following sentence to the main text:

“While the reflectivity of endothelium-aqueous interface is expected to be 100 times less than air-tear film interface (based on the refractive index difference), FFOCT images show only 10 times difference in brightness, because FFOCT acquires the images of amplitude (due to 2-phase amplitude retrieval

process) and not intensity like the conventional OCT. It should be noted that in the figures these interfaces appear with similar brightness due to adjustment of contrast and normalization.”